# FUNDAMENTAL LIMITATIONS ON SUBQUADRATIC ALTERNATIVES TO TRANSFORMERS

**Josh Alman**[*]
Department of Computer Science
Columbia University
New York, NY 10027, USA
`josh@cs.columbia.edu`

**Hantao Yu**[*]
Department of Computer Science
Columbia University
New York, NY 10027, USA
`hantao.yu@columbia.edu`

## ABSTRACT

The Transformer architecture is widely deployed in many popular and impactful Large Language Models. At its core is the attention mechanism for calculating correlations between pairs of tokens. Performing an attention computation takes quadratic time in the input size, and had become the time bottleneck for transformer operations. In order to circumvent this, researchers have used a variety of approaches, including designing heuristic algorithms for performing attention computations faster, and proposing alternatives to the attention mechanism which can be computed more quickly. For instance, state space models such as Mamba were designed to replace attention with an almost linear time alternative.

In this paper, we prove that any such approach cannot perform important tasks that Transformer is able to perform (assuming a popular conjecture from fine-grained complexity theory). We focus on document similarity tasks, where one is given as input many documents and would like to find a pair which is (approximately) the most similar. We prove that Transformer is able to perform this task, and we prove that this task cannot be performed in truly subquadratic time by any algorithm. Thus, any model which can be evaluated in subquadratic time – whether because of subquadratic-time heuristics for attention, faster attention replacements like Mamba, or any other reason – cannot perform this task. In other words, in order to perform tasks that (implicitly or explicitly) involve document similarity, one may as well use Transformer and cannot avoid its quadratic running time.

## 1 INTRODUCTION

The Transformer architecture (Vaswani et al., 2017) is widely used for natural language processing (Devlin et al., 2019b; Yang et al., 2019), computer vision (Dosovitskiy et al., 2021; Carion et al., 2020), and many other tasks, and has achieved state-of-the-art performance for numerous applications. At the core of the architecture is the attention mechanism which is designed to calculate the correlation between all pairs of tokens in a given input sequence. Namely, let $Q \in \mathbb{R}^{d_{\text{in}} \times m}$ be the query matrix, $K \in \mathbb{R}^{d_{\text{in}} \times m}$ be the key matrix and $V \in \mathbb{R}^{d_{\text{in}} \times d_{\text{out}}}$ be the value matrix. Given $X \in \mathbb{R}^{n \times d_{\text{in}}}$, an attention mechanism computes

$$A_{Q,K,V}(X) := \text{softmax}(XQK^\top X^\top)XV$$

where the softmax operator

$$\text{softmax}(v) = \frac{(\exp(v[1]), \ldots, \exp(v[n]))}{\sum_{i=1}^{n} \exp(v[i])}$$

for $v \in \mathbb{R}^n$ is applied to matrices row-wise. Computing the attention by straightforwardly following the definition above requires quadratic time (in the sequence length $n$), which prohibits efficient model training when the sequence length is too large. As a result, much effort has been devoted to overcoming this obstacle in recent years, and there are two major lines of research to tackle the problem.

---

[*] * denotes equal contribution

The first line of research argues that instead of computing attention exactly in the worst case, it often suffices to use heuristics which work well when the input data has additional structure, or to only return coarse approximations of the attention mechanism which can be computed more quickly. Examples including KDEformer (Zandieh et al., 2023), Reformer (Kitaev et al., 2020), Hyperattention (Han et al., 2024), Linformer (Wang et al., 2020), SMYRF (Daras et al., 2020), and Performer (Choromanski et al., 2021). In many cases, these heuristics result in algorithms which run in close to linear time. However, these techniques usually have corresponding downsides, such as model accuracy drops, or performance gains which do not appear to scale to large inputs.

The second line of research argues that instead of computing or approximating attention, we can replace the standard attention mechanism with new, different mechanisms which can be computed faster. Models such as Longformer (Beltagy et al., 2020), Synthesizer (Tay et al., 2021), Routing transformers (Roy et al., 2021), and MAMBA (Gu & Dao, 2023) all aim to circumvent the quadratic barrier by proposing new attention alternatives. A priori, these techniques would result in weaker expressiveness since they replace attention's calculation of token interactions with simpler alternatives, although most also provide empirical evidence that the loss in accuracy at certain tasks is not large.

In this paper, we prove that any approach that takes subquadratic time, no matter whether it uses heuristic or approximations, or a new architecture, or a completely different approach, is inherently unable to perform important learning tasks that a transformer is able to perform. By using tools and popular hardness conjectures from the area of fine-grained complexity theory, we show that many learning tasks involving document similarity cannot possibly be solved in subquadratic time using *any* algorithmic approach. This implies that subquadratic alternatives to standard transformers are not able to solve these simple and natural tasks in machine learning and NLP. To complement this, we show that standard transformers (even simple transformers with one layer and one attention head) are able to perform these tasks, thereby establishing a separation between standard transformers and these new approaches. In other words, we prove that accuracy loss for any task relating to document similarity is *unavoidable* for any subquadratic approach, even when compared only to the simplest transformers, because of the inherent computational complexity of the task.

## 1.1 DOCUMENT SIMILARITY

In this work we will be focusing on document similarity tasks. We will show that standard transformers are capable of solving these tasks but subquadratic alternatives to transformers cannot.

Document similarity is a fundamental area in natural language processing with many applications including recommender systems (Ostendorff, 2020), search engines (Mahdi et al., 2018), and plagiarism-detection (Baba et al., 2017). For a given document $D$, we first need to define a document embedding to transform it into a vector $v \in \mathbb{R}^d$, and we will then measure how similar two documents are by using a similarity measure on their embedding vectors. There are many ways to embed a document as a vector including Doc2Vec (Le & Mikolov, 2014), TF-IDF (Sparck Jones, 1988), BERT (Devlin et al., 2019b), bag-of-words (Harris, 1954), and many ways to measure how similar two documents are including cosine similarity, Euclidean distance, and Jaccard Similarity. In this work, we focus on two of the most popular options, bag-of-words embedding and cosine similarity, although our results extend naturally to almost any reasonable alternatives.

**Bag-of-words Embedding.** Bag-of-words embedding is a well-studied method of embedding (Blei et al., 2003) that is commonly used for text classification (Jin et al., 2016), radiology (Juluru et al., 2021) and many other settings, especially in NLP. Given a document $D$ and a list of $\ell$ key words, the bag-of-words embedding of $D$ is a vector $v \in \{0, 1\}^\ell$ such that the $i$-th entry of $v$ corresponds to whether the $i$-th key word exists in $D$ or not. [1] For example, if a document only contains one sentence "*There are ten apples on the apple tree*" and the key words chosen are "*apple*", "*tree*", "*computer*" and "*ten*", this document will have bag-of-words embedding $(1, 1, 0, 1)$.

**Cosine Similarity.** Cosine similarity is one of the most commonly used method to measure how similar two documents are. Given two document embeddings $v, w \in \mathbb{R}^d$, the cosine similarity is

---

[1]Sometimes the entries are also frequency counts, but that would only make the embedding vectors more complex. Since our main goal is to show hardness, we will focus on the simpler binary case.

given by

$$\frac{\langle v, w \rangle}{\|v\|_2 \cdot \|w\|_2} \in [0, 1]$$

where 1 represents complete similarity and 0 represents no similarity. It is defined in this way, rather than just taking the inner product $\langle v, w \rangle$, so that two documents (vectors) with large magnitude can still be considered close if they have similar directions. Cosine similarity is one of the most popular and effective measures; for instance, Sanchez-Gomez et al. (2021) found that when used for extractive multi-document text summarization, cosine similarity gives the best results.

**Problem Statement.** We define our most similar document (MSD) and least similar document (LSD) problems as: given a set of $n$ binary vectors $v_1, \ldots, v_n$ of length $d$ (document embeddings), the goal is to find two documents that are the most/least similar to each other in terms of cosine similarity. There are many natural variants of these two problems, and we prove similar hardness for all of them:

1. (Bichromatic MSD, LSD) Sometimes we have two sets of documents $A, B$ and we want to find one document from each set such that the pair is (un)similar.

2. ($\gamma$-MSD, $\gamma$-LSD) Sometimes we might only need to find a pair of documents that is *approximately* the most (un)similar, (up to an approximation factor $\gamma$) and not necessarily the optimal pair.

3. ($\mathsf{MSD}_{n,d,t}$, $\mathsf{LSD}_{n,d,t}$) Sometimes we only want to know if there exists a pair whose cosine similarity is above (or below) a threshold $t \in [0, 1]$.

These variants occur in many practical scenarios when using a large language model. They can arise explicitly when the descriptions of $n$ documents of size $\ell$ are given to a language model, and the model is asked to find the most similar pair of documents. However, there are many scenarios where document similarity can arise implicitly as well, such as in plagiarism detection and team matching.

## 1.2 MAIN RESULTS

Our hardness results are based on a prevalent conjecture from fine-grained complexity theory called the Strong Exponential Time Hypothesis ($\mathsf{SETH}$):

*For every $\varepsilon > 0$, there is an integer $k$ such that $k\mathsf{SAT}$ with $n$ variables requires $\Omega(2^{(1-\varepsilon)n})$ time.*

$\mathsf{SETH}$ was first introduced by Impagliazzo & Paturi (2001), and is a popular strengthening of the conjecture that $\mathsf{P} \neq \mathsf{NP}$. (In other words, proving that $\mathsf{SETH}$ is true implies $\mathsf{P} \neq \mathsf{NP}$.) Since then, there has been a long line of work studying and making use of $\mathsf{SETH}$. Prior work has given theoretical evidence for $\mathsf{SETH}$ (Impagliazzo & Paturi, 2001; Abboud et al., 2018; Vassilevska Williams, 2015), and has used $\mathsf{SETH}$ to prove hardness of problems in many different areas of algorithm design. See the survey Williams (2018) for a detailed background.

**Main Results: Limitations of subquadratic alternatives.** We show that $\mathsf{MSD}, \mathsf{LSD}$ and their variants require quadratic time assuming $\mathsf{SETH}$ for some natural choice of parameters, and therefore any subquadratic alternatives to transformers are not able to solve them due to computational constraints. The formal hardness results are as follows, and vary slightly in the dimension parameter $\ell$ depending on the details of the problem:

**Theorem 1.1** (Theorem 3.1 and Corollary 3.2). *Assuming $\mathsf{SETH}$, for every $\varepsilon > 0$, there exists a constant $c > 0$ such that $\mathsf{LSD}_{n,\ell}$ cannot be solved in $O(n^{2-\varepsilon})$ time when $\ell = c \log n$. Moreover, the same lower bound also holds for $\mathsf{LSD}_{n,\ell,t}$ for some $0 < t < 1$, $\gamma$-$\mathsf{LSD}_{n,\ell}$ for any $\gamma \geq 1$, and bichromatic $\mathsf{LSD}_{n,\ell}$.*

**Theorem 1.2** (Theorem 3.3 and Corollary 3.4). *Assuming $\mathsf{SETH}$, for every $\varepsilon > 0$, there exists a constant $c > 0$ such that $\mathsf{MSD}_{n,\ell}$ cannot be solved in $O(n^{2-\varepsilon})$ time when $\ell = n^{\frac{c}{\log \log n}}$. Moreover, the same lower bound also holds for $\mathsf{MSD}_{n,\ell,t}$ for some $0 < t < 1$ and $\gamma$-$\mathsf{MSD}_{n,\ell}$ for any $1 \leq \gamma \leq \text{polylog}(n)$.*

For bichromatic MSD we can obtain a stronger hardness result.

**Theorem 1.3** (Theorem 3.3 and Corollary 3.4). *Assuming* SETH, *for every* $\varepsilon > 0$, *there exists a constant* $c > 0$ *such that bichromatic* $\mathsf{MSD}_{n,\ell}$ *cannot be solved in* $O(n^{2-\varepsilon})$ *time when* $\ell = c \log n$.

In all these problems, we prove hardness when $\ell$ is $\Theta(\log n)$ or $n^{o(1)}$. This is a natural choice: for any smaller $\ell < \log n$, there could be at most $2^\ell < n$ vectors under a bag-of-words embedding, which means that there would be duplicate vectors in our instance. Making $\ell$ larger can only make the problem harder.

It follows that any language task which, either explicitly or implicitly, involves solving any of these document similarity problems, cannot be solved in subquadratic time *when the input size is large enough*, no matter what the parameters or architecture of the language model are.[2]

**Main Results: Representational strength of standard transformers.**   When a problem requires quadratic time to solve, this means that subquadratic-time models cannot solve it, but it does not necessarily mean that a transformer model can solve it. For example, Sanford et al. (2024b) defined a problem called "Match3" which can be solved in quadratic time by a textbook algorithm, but which they prove cannot be solved by a one-layer transformer unless it has a lot of attention heads or a very high embedding dimension.

We show that this is not the case for MSD and LSD by showing that a single standard attention unit with input and output MLPs can solve $\mathsf{MSD}_{n,d,t}$, $\mathsf{LSD}_{n,d,t}$ and a simpler version of MSD, the Orthogonal Vectors problem (OV), where one is given a set of binary vectors and needs to determine if there exists a pair of vectors that are orthogonal. Thus, these problems establish a separation between standard transformers and subquadratic alternatives to transformers.

**Theorem 1.4** (Theorems 4.1 and C.2). *A single unit of standard attention with input and output MLPs, embedding dimension* $\ell + 1$ *can solve* $\mathsf{OV}_{n,\ell}$ *and* $\mathsf{MSD}_{n,\ell,t}$, $\mathsf{LSD}_{n,\ell,t}$ *for any* $0 \le t \le 1$.

In principle, there could be concerns with representational results like these that the weights of the model are complicated and hard to find in training. However, our constructions of transformers that solve these problems are very simple: our MLPs are piece-wise linear functions that are easy to approximate/compute, and our key/query/value matrices in the attention unit are also simple, sparse and low-rank matrices with small entries.

## 1.3   RELATED WORK

In recent years, several theoretical and algorithmic aspects of transformers have been extensively studied. We discuss next two aspects that are most relevant to our work, and which we build on in the proofs of our results.

**Representational strengths of Transformers.**   Representational strengths of transformers have been widely studied in recent years. It has been shown that transformers have several natural limitations, including not being able to model periodic finite-state languages or hierarchical structure (Hahn, 2020), and not being able to recognize some counter languages without large depth (Bhattamishra et al., 2020). On the other hand, transformers are able to recognize formal languages such as Dyck languages (Yao et al., 2021), simulate finite-state automata (Liu et al., 2023) with $O(\log n)$ depth, and simulate Turing machines if given enough depth (Wei et al., 2024; Merrill & Sabharwal, 2024). There is also a line of work (Hao et al., 2022; Merrill et al., 2022) that understands what transformers can compute through the lens of circuit complexity; see Strobl et al. (2024) for a comprehensive survey.

Another line of work has shown that transformers can compute particular problems of interest, including PARITY (which perceptrons are notably unable to compute) (Chiang & Cholak, 2022) and learning problems that attention is particularly suited toward like "sparse averaging" and "$k$-hop induction heads" (Sanford et al., 2024a;b).

---

[2]We briefly emphasize that the parameter $\ell$ in these similarity problems need not be related to the architecture or parameters (like $d_{\text{in}}, d_{\text{out}}$, etc) of a language model which solves the problems. For instance, to ask a language model to solve $\mathsf{MSD}_{n,\ell}$, we may ask it "Which of the following paragraphs is most similar?" followed by a list of $n$ different paragraphs of at most $\ell$ words each. Thus, the input to the language model would be a string of length $O(n\ell)$, and since $\ell < n^{o(1)}$ is small compared to $n$, our result shows that a language model would need to take quadratic time in the string length to answer this type of query.

**Faster attention mechanisms.** As previously discussed, attention computation remains a bottleneck for efficiency, and many different approaches have been proposed to tackle this issue. These approaches typically result in accuracy loss (and a consequence of our main result is theoretical proof that this is necessary), which has mostly been investigated empirically.

The main prior work on theoretical limitations of subquadratic transformers we're aware of is Sanford et al. (2024a). Among other results, they study "kernel-based subquadratic attention" in which one heuristically computes attention faster by approximating intermediate matrices in the attention computation either by sparse matrices (Kitaev et al., 2020; Roy et al., 2021; Daras et al., 2020) or low-rank matrices (Choromanski et al., 2021; Katharopoulos et al., 2020). Sanford et al. (2024a) defined a "$k$-hop induction heads" task and proved that transformers can perform this task but kernel-based subquadratic attention models cannot. Our limitation result is more general than this, applying to any approach that runs in subquadratic time, rather than needing to focus on a particular architecture or heuristic.

**Fine-Grained Complexity and Machine Learning** Fine-grained complexity theory has been successful at proving conditional lower bounds for problems in diverse areas of algorithm design, such as in graph theory (Abboud & Williams, 2014; Abboud et al., 2015a; Williams & Williams, 2018) and combinatorial optimization (Rubinstein, 2018; Alman et al., 2024; Künnemann et al., 2017; Backurs & Indyk, 2015). See Williams (2018) for a detailed survey.

Recently, it has been shown that many problems in machine learning are also inherently hard assuming popular conjectures in fine-grained complexity. For example, Backurs et al. (2017); Alman & Guan (2024) use SETH to give a lower bound on the time to perform kernel density estimation, Hu et al. (2024a) use SETH to demonstrate a computational phase transition in modern Hopfield models, Hu et al. (2024c) and Hu et al. (2024b) use SETH to characterize the computational limits of diffusion transformers and Low-Rank Adaptation for transformers respectively, and Duman Keles et al. (2023); Alman & Song (2024) use SETH to give a lower bound on computational complexity of approximating the attention mechanism itself. We remark that our results, together, give a new, alternate proof of the hardness of attention assuming SETH. Indeed, we prove that a single attention unit can solve MSD, and that MSD requires quadratic time assuming SETH, which together imply that evaluating the attention unit requires quadratic time.

**Hardness of similarity search.** Similarity search has been a fundamental area in modern machine learning, and the efficiency of similarity search algorithms has been well-studied through the lens of fine-grained complexity. Perhaps the most well-known problem in this area is the nearest neighbor problem.

Nearest Neighbor is a fundamental problem in machine learning which has been the subject of decades of research (Indyk & Motwani, 1998; Andoni & Indyk, 2008; Andoni & Razenshteyn, 2015; Andoni et al., 2017; Tao et al., 2002; Engels et al., 2024; Uddin et al., 2022; Gu et al., 2019). Given a dataset $P \subseteq \mathbb{R}^d$ with $n$ points, we want to preprocess it such that given a query point $q \in \mathbb{R}^d$, one can find its nearest neighbor in $P$ (in some metric) efficiently. The nearest neighbor problem gives one common way of classifying objects in machine learning: given an object with unknown label, one just find its nearest neighbor in the dataset and use its label as the label for the target object. In addition, it has many applications in classical similarity search over different types of data including text, images, audio (see Shakhnarovich et al. (2006) for a complete overview).

There are many natural variants of the nearest neighbor problem, including the closest (furthest) pair problem where one is given a dataset $P$ and wants to find the two points from $P$ that are the closest (furthest). This problem is exactly Min-IP(Max-IP) if we let inner product be the measure of closeness. Under standard Euclidean distance, one can also express the distance between two points as $\|p - q\| = \|p\| + \|q\| - 2\langle p, q \rangle$ such that when all points have the same $\ell_2$ norm, finding the closest pair is equivalent to finding the pair with the largest inner product, i.e. Max-IP. In fact, it has been shown that assuming SETH is true, finding closest pairs in Euclidean or Manhattan distance both require quadratic time (Alman & Williams, 2015; Rubinstein, 2018). We will use this in our proofs below.

## 2 PRELIMINARIES

**Notation.** For a vector $v$, we use $v[i]$ to denote its $i$-th entry for all $i$. For a matrix $A$, we use $A_{i,:}$ to denote the $i$-th row of $A$ and $A_{:,j}$ to denote the $j$-th column of $A$. Given a positive integer $d$, we use $\mathbf{1}_d \in \mathbb{R}^d$ to denote the vector whose entries are all 1. Given two vectors $v \in \mathbb{R}^a$, $w \in \mathbb{R}^b$, we use $v \otimes w \in \mathbb{R}^{ab}$ to denote the Kronecker product of $v$ and $w$ (whose entries are all the products of an entry of $v$ and an entry of $w$) and $v \circ w \in \mathbb{R}^{a+b}$ to denote the concatenation of $v$ and $w$. $\| \cdot \|$ refers to $\ell_2$ norm unless otherwise specified. Given a binary vector $v \in \{0,1\}^d$, we use $\bar{v} \in \{0,1\}^d$ to denote the vector where all entries are flipped. The $\mathrm{softmax}$ operator, when given a vector $v \in \mathbb{R}^n$, outputs a vector in $\mathbb{R}^n$ given by

$$\mathrm{softmax}(v) = \frac{(\exp(v[1]), \ldots, \exp(v[n]))}{\sum_{i=1}^n \exp(v[i])}.$$

For matrices $A \in \mathbb{R}^{n \times n}$, we apply $\mathrm{softmax}$ operator row-wise, so that

$$\mathrm{softmax}(A)_{i,:} = \mathrm{softmax}(A_{i,:}).$$

### 2.1 THE TRANSFORMER ARCHITECTURE

Transformer is a machine learning architecture composed mainly of attention layers and multi-layer perceptrons (MLP). We model the input to a attention unit is a $n \times d_{\mathrm{in}}$ matrix where $d_{\mathrm{in}}$ is the input dimension and the output of a attention is a $n \times d_{\mathrm{out}}$ matrix where $d_{\mathrm{out}}$ is the output dimension.

**Definition 2.1** (attention). *For input dimension $d_{\mathrm{in}} \in \mathbb{N}$, output dimension $d_{\mathrm{out}} \in \mathbb{N}$, embedding dimension $m \in \mathbb{N}$, matrices $Q, K \in \mathbb{R}^{d_{\mathrm{in}} \times m}$ and $V \in \mathbb{R}^{d_{\mathrm{in}} \times d_{\mathrm{out}}}$, a* attention *is a mapping $A_{Q,K,V} : \mathbb{R}^{n \times d_{\mathrm{in}}} \to \mathbb{R}^{n \times d_{\mathrm{out}}}$ by*

$$A_{Q,K,V}(X) = \mathrm{softmax}(XQK^\top X^\top)XV.$$

*We use $\mathcal{A}_{d_{\mathrm{in}}, m, d_{\mathrm{out}}} = \{A_{Q,K,V} : Q \in \mathbb{R}^{d_{\mathrm{in}} \times m}, K \in \mathbb{R}^{d_{\mathrm{in}} \times m}, V \in \mathbb{R}^{d_{\mathrm{in}} \times d_{\mathrm{out}}}\}$ to denote all such* attentions.

An attention layer consists of many attentions in parallel. Upon receiving input $X$, each attention computes an output locally, then the results are all concatenated into a large matrix before being sent to the next layer. Our constructions in this paper will only need transformers with one layer and one single unit of attention to illustrate representational strength (transformers with more layers and heads could only be stronger), so we omit the formal definitions of attention layers.

A *multi-layer perceptron* (MLP) is a type of neural network that is used to learn nonlinear relationships in data. Mathematically, it is usually formulated as a neural network with different types of activation functions (Bartlett et al., 2017; Montúfar et al., 2014; Jacot et al., 2018) or sometimes as a more specific threshold circuit (Maass et al., 1994). Since the universal approximation theorem (Hornik et al., 1989) states that any continuous function with a finite support can be approximated by a neural network with one hidden layer, Sanford et al. (2024a;b) modeled MLP as an arbitrary function $\phi : \mathbb{R}^d \to \mathbb{R}^{d'}$ defined on fixed-precision vectors, and we will notationally use that definition here.

**Definition 2.2** (Multi-player perceptron). *A multi-layer perceptron is represented by some continuous function $\varphi : \mathbb{R}^a \to \mathbb{R}^b$ for some positive integers $a, b$. We can apply $\varphi$ to a matrix row-wisely: given any matrix $X \in \mathbb{R}^{n \times a}$, $\varphi(X) = (\varphi(X_1), \ldots, \varphi(X_n)) \in \mathbb{R}^{n \times b}$.*

That said, in all our constructions in this paper, it will suffice to use MLPs which are simple, piecewise-linear functions that fit in all the categories discussed above, and can be easily approximated by a small neural network or any other MLP definition.

In this work, our transformer (with a single attention unit) will be defined as a composition of the first MLP, then one attention unit, then the second MLP. This is a natural model for many well-known transformer models including BERT (Devlin et al., 2019a), GPT-3 Brown et al. (2020), GPT-4 (OpenAI, 2023) and is typically used in theoretical work on simple transformers (Phuong & Hutter, 2022; Sanford et al., 2024a;b).

Recall in particular that we will be designing such transformers for document similarity problems. In this case, given $n$ documents $D_1, \ldots, D_n$, we will let their bag-of-words embeddings

$$(\mathsf{BOW}(D_1), \ldots, \mathsf{BOW}(D_n))^\top \in \mathbb{R}^{n \times d}$$

be the input to our transformer, and the output of the transformer will be a real number indicating the answer to our problems.

**Definition 2.3.** *A* transformer *is a mapping* $\mathsf{TF} : \mathbb{R}^{n \times d} \to \mathbb{R}$ *specified by a attention unit* $A_{Q,K,V}$ *and two MLPs* $\varphi_1 : \mathbb{R}^{n \times d} \to \mathbb{R}^{n \times d_{\mathrm{in}}}, \varphi_2 : \mathbb{R}^{n \times d_{\mathrm{out}}} \to \mathbb{R}$. *Upon an embedding matrix* $E \in \mathbb{R}^{n \times d}$, *the transformer outputs* $\varphi_2(A_{Q,K,V}(\varphi_1(E)))$.

To emphasize, this is a very simplified model of a transformer with a single attention unit. We say that a transformer $\mathsf{TF}$ solves a problem whose input is a matrix $E \in \mathbb{R}^{n \times d}$ if $\mathsf{TF}(E)$ is the answer of the problem on instance $E$. For example, for decision problems like $\mathsf{MSD}_{n,d,t}$, we say $\mathsf{TF}$ solves $\mathsf{MSD}_{n,d,t}$ if for all input $v_1, \ldots, v_n$ and $E$ such that $E_{i,:} = v_i$ for all $i$, $\mathsf{TF}(E) = 1$ if there exists a pair $\frac{\langle v_i, v_j \rangle}{\|v_i\| \cdot \|v_j\|} \geq t$ and $\mathsf{TF}(E) = 0$ otherwise.

## 2.2 Fine-Grained Complexity

We first introduce some common notions from fine-grained complexity. Many proofs in this section are deferred to Appendix A, and we also refer the reader to Appendix A for a more detailed introduction to fine-grained complexity.

In fine-grained complexity, one is usually interested in whether we can improve the running time of our algorithms by a polynomial factor. For example, the $\mathsf{OV}$ problem (defined below) has a straightforward quadratic time (ignoring logarithmic factors) algorithm, and it is a major open problem to determine whether there exists a faster, $O(n^{1.99})$ time algorithm. We say that an algorithm is *truly subquadratic* [3] if it runs in time $O(n^{2-\varepsilon})$ for some constant $\varepsilon > 0$. In this work, all our problems have quadratic time solutions, and we are interested in whether truly subquadratic time algorithms exist.

A key technique (that we will use frequently in this work) in fine-grained complexity is the *fine-grained reduction*, which is a way to connect the running times of different problems. If problems $\mathcal{P}$ and $\mathcal{Q}$ both have quadratic time algorithms, we say that $\mathcal{P}$ *reduces to* $\mathcal{Q}$ (sometimes we would also say $\mathcal{P}$ is easier than $\mathcal{Q}$ or $\mathcal{Q}$ is harder than $\mathcal{P}$) if a truly subquadratic time algorithm for $\mathcal{Q}$ implies a truly subquadratic time algorithm for $\mathcal{P}$. We say $\mathcal{P}$ and $\mathcal{Q}$ are *subquadratic equivalent* if they reduce to each other, i.e. there is a truly subquadratic time algorithm for $\mathcal{P}$ if and only if there is a truly subquadratic time algorithm for $\mathcal{Q}$. Such relationships are proved by careful reductions, and we will see many examples soon.

### 2.2.1 Hardness Conjectures: SETH and OVC

We restate our central hardness conjecture here.

**Definition 2.4** (Strong Exponential Time Hypothesis (SETH)). *For any* $\varepsilon > 0$, *there exists a positive integer* $k$ *such that* $k\mathsf{SAT}$ *requires* $\Omega(2^{(1-\varepsilon)n})$ *time, where* $n$ *is the number of variables in the CNF.*

SETH has been one of the biggest open problem in fine-grained complexity, and one of the reasons is that SETH being true would imply $\mathsf{P} \neq \mathsf{NP}$. See Section 1.2 for a detailed explanation of its significance and why many people believe that it is true.

We also introduce the Orthogonal Vectors problem, which is an important problem in fine-grained complexity and will be a key intermediate problem in some of our proofs.

**Definition 2.5** (Orthogonal Vectors ($\mathsf{OV}_{n,\ell}$)). *Given binary vectors* $v_1, \ldots, v_n \in \{0,1\}^\ell$, $\mathsf{OV}_{n,\ell}$ *asks to determine if there exists a pair* $i \neq j$ *such that* $\langle v_i, v_j \rangle = 0$.

Much effort has been made to give a truly subquadratic algorithm for $\mathsf{OV}_{n,c \log n}$ for all $c$, but none has succeeded. Therefore, Williams (2005) proposed the Orthogonal Vectors Conjecture, which asserts that such an algorithm does not exist.

---

[3] Usually in fine-grained complexity, a subquadratic time algorithm means the algorithm runs in time $o(n^2)$, and a truly subquadratic time algorithm means the algorithm runs in time $O(n^{2-\varepsilon})$ for a fixed constant $\varepsilon > 0$. For example, $O(n^2/\log n)$ is subquadratic but not truly subquadratic. By contrast, in the context of fast attention, prior machine learning literature often just calls an approach subquadratic to mean that it is truly subquadratic, or often even almost linear time. In this work, we will use the fine-grained complexity definition and refer to such approaches as truly subquadratic.

**Conjecture 2.6** (OVC). *For any $\varepsilon > 0$, there exists a constant $c > 0$ such that $\mathsf{OV}_{n,c\log n}$ cannot be solved in $O(n^{2-\varepsilon})$ time.*

Williams (2005) showed that assuming SETH is true, then OVC is true (the other direction is unknown). Our paper will use these two conjectures interchangeably such that our hardness results can be obtained from either conjecture.

There is also a bichromatic version of $\mathsf{OV}_{n,\ell}$ where one is given two sets of vectors $A = \{a_1, \ldots, a_n\}, B = \{b_1, \ldots, b_n\}$ such that $a_i, b_j \in \{0,1\}^\ell$ and wants to determine if there exists $i, j$ such that $\langle a_i, b_j \rangle = 0$. In fact, these two problems are subquadratic equivalent (see Theorem A.4 for proof).

### 2.2.2 MINIMUM INNER PRODUCT

In this section we introduce the minimum inner product problem, an important problem related to similarity search.

**Definition 2.7** (Min-IP). *Given a set of binary vectors $v_1, \ldots, v_n \in \{0,1\}^\ell$, $\mathsf{Min\text{-}IP}_{n,\ell}$ asks to find one pair of $1 \le i, j \le n, i \ne j$ such that $\langle v_i, v_j \rangle$ is minimum.*

Sometimes we are happy with finding a pair of vectors whose inner product is close enough to optimal, so we also introduce the approximate Min-IP problem as follows.

**Definition 2.8** ($\gamma$-Min-IP). *Given a set of binary vectors $v_1, \ldots, v_n \in \{0,1\}^\ell$, $\gamma$-$\mathsf{Min\text{-}IP}_{n,\ell}$ asks to find one pair of $1 \le i, j \le n, i \ne j$ such that $\langle v_i, v_j \rangle$ is a $\gamma$-approximation of the minimal inner product.*

It is not hard to see that Min-IP and $\gamma$-Min-IP are both at least as hard as OV for any $\gamma \ge 1$ (just find the minimum inner product and see if it is $0$, and any multiplicative approximation of $0$ must be $0$). Therefore, assuming OVC, for any $\varepsilon > 0$ there exists $c > 0$ such that $\mathsf{Min\text{-}IP}_{n,c\log n}$ cannot be solved in $O(n^{2-\varepsilon})$ time.

In addition, there is a decision version of Min-IP, by which we denote $\mathsf{Min\text{-}IP}_{n,\ell,t}$, where one wants to know whether there exists a pair of vectors whose inner product is at most $t$ for some $0 \le t \le \ell$.

**Definition 2.9** (Min-IP decision version). *Given a set of binary vectors $v_1, \ldots, v_n \in \{0,1\}^\ell$ and $0 \le t \le \ell$, $\mathsf{Min\text{-}IP}_{n,\ell,t}$ asks to determine if there exists one pair of $1 \le i, j \le n, i \ne j$ such that $\langle v_i, v_j \rangle \le t$.*

The bichromatic versions of these problems can be defined analogously: given two sets $A, B$ with vectors in $\{0,1\}^\ell$, one needs to find $i, j$ that achieves (for bichromatic $\mathsf{Min\text{-}IP}_{n,\ell}$) or approximates (for bichromatic $\gamma$-$\mathsf{Min\text{-}IP}_{n,\ell}$) the minimal $\langle a_i, b_j \rangle$. One can obtain a truly subquadratic algorithm for all three problems above given a truly subquadratic algorithm for their bichromatic versions (see Theorem A.8 for proof).

### 2.2.3 MAXIMUM INNER PRODUCT

One can analogously define Max-IP and its variants; see Appendix A.3 for the formal definitions.

It is less obvious whether Max-IP is a harder problem than OV or not. The answer is positive, see Theorem A.13 for a simple proof.

In fact, Karthik & Manurangsi (2020) proved a stronger statement which says that even approximate $\mathsf{Max\text{-}IP}_{n,\ell}$ is harder than $\mathsf{OV}_{n,\ell}$ for some approximation factor.

### 2.3 DOCUMENT SIMILARITY PROBLEMS

In this section we formally define the MSD, LSD problems that we will study. First we define the MSD variants, which are defined similarly to Max-IP variants.

**Definition 2.10** (MSD). *Given $n$ document embeddings $v_1, \ldots, v_n \in \{0,1\}^\ell$, $\mathsf{MSD}_{n,\ell}$ asks to find $1 \le i, j \le n, i \ne j$ such that $\frac{\langle v_i, v_j \rangle}{\|v_i\| \cdot \|v_j\|}$ is the maximum.* [4]

---

[4]In all versions of MSD and LSD, we assume that there are no zero vectors.

Even though MSD looks similar to Max-IP, notice that they are not the same problem because of normalization. For example, $v = (1, 1, \dots, 1) \in \mathbb{R}^{\ell}$ and $w = (1, 1, \dots, 1, 0, \dots, 0) \in \mathbb{R}^{\ell}$ where $w$ has $\ell/2$ ones have a very large inner product but they might not be considered similar. In contrast, $v' = (1, 1, \dots, 1, 0, \dots, 0) \in \mathbb{R}^{\ell}$ where $v'$ has 10 ones and $w' = (0, 1, \dots, 1, 0, \dots, 0) \in \mathbb{R}^{\ell}$ where $w'$ has 10 ones have a inner product of 9 but they are very similar in terms of cosine similarity.

**Definition 2.11** ($\gamma$-MSD). *Given $n$ document embeddings $v_1, \dots, v_n \in \{0, 1\}^{\ell}$, $\gamma$-MSD$_{n,\ell}$ asks to find $1 \leq i^*, j^* \leq n, i^* \neq j^*$ such that*

$$\frac{1}{\gamma} \cdot \max_{1 \leq i,j \leq n} \frac{\langle v_i, v_j \rangle}{\|v_i\| \cdot \|v_j\|} \leq \frac{\langle v_{i^*}, v_{j^*} \rangle}{\|v_{i^*}\| \cdot \|v_{j^*}\|} \leq \max_{1 \leq i,j \leq n} \frac{\langle v_i, v_j \rangle}{\|v_i\| \cdot \|v_j\|}.$$

**Definition 2.12** (MSD decision version). *Given $n$ document embeddings $v_1, \dots, v_n \in \{0, 1\}^{\ell}$, MSD$_{n,\ell,t}$ asks to determine if there exists $1 \leq i, j \leq n, i \neq j$ such that $\frac{\langle v_i, v_j \rangle}{\|v_i\| \cdot \|v_j\|} \geq t$.*

It is not hard to see that $\gamma$-MSD$_{n,\ell}$ and MSD$_{n,\ell,t}$ are both easier than MSD. In addition, notice that the number of possible $t$ can be considered as discrete because there could only be $O(\ell^3)$ possible values of $\frac{\langle v_i, v_j \rangle}{\|v_i\| \cdot \|v_j\|}$. As a result, the existence of truly subquadratic time algorithm for MSD$_{n,\ell,t}$ for all $t \in [0, 1]$ would imply a truly subquadratic time algorithm for MSD$_{n,\ell}$ using binary search.

Bichromatic versions of these problems can be defined analogously and the proof of Lemma A.8 again tells us that bichromatic versions are harder.

One can analogously define LSD and its variants; see Appendix A.4 for the formal definitions.

# 3 HARDNESS OF DOCUMENT SIMILARITY

In this section, we show that assuming SETH or OVC, for any $\varepsilon > 0$, there exists a constant $c > 0$ (only depends on $\varepsilon$) such that many variants of LSD$_{n,c \log n}$, MSD$_{n,c \log n}$ require $O(n^{2-\varepsilon})$ time.

**Theorem 3.1.** *Assuming SETH or OVC, for every $\varepsilon > 0$, there exists a constant $c > 0$ such that $\gamma$-LSD$_{n,\ell}$ cannot be solved in $O(n^{2-\varepsilon})$ time for any $\gamma \geq 1$ when $\ell = c \log n$.*

*Proof.* Assume by contradiction that there exists an algorithm $\mathcal{A}$ for $\gamma$-LSD$_{n,c \log n}$ that runs in time $O(n^{2-\varepsilon})$ for some $\gamma \geq 1, \varepsilon > 0$ and any constant $c > 0$. We show that OV$_{n,c \log n}$ can be solved in time $O(n^{2-\varepsilon})$ for any constant $c > 0$, which refutes OVC and SETH.

Given vectors $v_1, \dots, v_n \in \{0, 1\}^{\ell}$ where $\ell = c \log n$ for any constant $c$, if any vector is the zero vector (we can check in time $O(n\ell)$), then output yes. Otherwise we run $\mathcal{A}$ on $v_1, \dots, v_n$ to compute $i^*, j^*$ such that

$$\min_{1 \leq i,j \leq n} \frac{\langle v_i, v_j \rangle}{\|v_i\| \cdot \|v_j\|} \leq \frac{\langle v_{i^*}, v_{j^*} \rangle}{\|v_{i^*}\| \cdot \|v_{j^*}\|} \leq \gamma \cdot \min_{1 \leq i,j \leq n} \frac{\langle v_i, v_j \rangle}{\|v_i\| \cdot \|v_j\|}.$$

Observe that $\min_{1 \leq i,j \leq n} \frac{\langle v_i, v_j \rangle}{\|v_i\| \cdot \|v_j\|} = 0$ if and only if there exists a pair of orthogonal vectors, which implies that there exists a pair of orthogonal vectors if and only if $\mathcal{A}$ outputs a pair of orthogonal vectors. The total amount of time needed for OV$_{n,\ell}$ is therefore $O(n\ell + n^{2-\varepsilon}) = O(n^{2-\varepsilon})$, which refutes SETH. $\qquad \square$

Since $\gamma$-LSD$_{n,\ell}$ is easier than LSD$_{n,\ell}$ and bichromatic $\gamma$-LSD$_{n,\ell}$, the same lower bound applies to these two problems as well. In addition, there must exist $t \in [0, 1]$ such that LSD$_{n,\ell,t}$ cannot be solved in $O(n^{2-\varepsilon})$ time when $\ell = c \log n$ because otherwise that would imply a $O(n^{2-\varepsilon})$ time algorithm for LSD$_{n,\ell}$ using binary search.

**Corollary 3.2.** *Assuming SETH or OVC, for every $\varepsilon > 0$, there exists a constant $c > 0$ such that LSD$_{n,d}$ cannot be solved in $O(n^{2-\varepsilon})$ time when $\ell \geq c \log n$. Moreover, the same lower bound holds for bichromatic $\gamma$-LSD$_{n,\ell}$ for all $\gamma \geq 1$ and LSD$_{n,\ell,t}$ for some $t \in [0, 1]$.*

A similar hardness result for $\gamma$-MSD$_{n,d}$ can be derived with much more complicated techniques. Our proof follows the same idea as Theorem 6.1 of Karthik & Manurangsi (2020) which uses graph constructions, and we delay the proof of Theorem 3.3 to Appendix B.

**Theorem 3.3.** *Assuming* SETH, *for every* $\varepsilon > 0$, *there exists a constant* $c > 0$ *such that* $\gamma$-$\mathsf{MSD}_{n,\ell}$ *cannot be solved in* $O(n^{2-\varepsilon})$ *time when*

$$\ell \geq (\log n)^{\frac{c \log n}{(\log \log n)^2}} \text{ and } \gamma \leq \left(1 + \frac{1}{\log \log n}\right)^{\frac{\log n}{(\log \log n)^2}}.$$

Similarly, the hardness result also applies to harder problems including $\mathsf{MSD}_{n,\ell}$ and bichromatic $\gamma$-$\mathsf{MSD}_{n,\ell}$.

**Corollary 3.4.** *Assuming* SETH *or* OVC, *for every* $\varepsilon > 0$, *there exists a constant* $c > 0$ *such that* $\mathsf{MSD}_{n,\ell}$ *cannot be solved in* $O(n^{2-\varepsilon})$ *time when* $\ell \geq (\log n)^{\frac{c \log n}{(\log \log n)^2}}$. *Moreover, the same lower bound holds for bichromatic* $\gamma$-$\mathsf{MSD}_{n,\ell}$ *for all* $1 \leq \gamma \leq (1 + \frac{1}{\log \log n})^{\frac{\log n}{(\log \log n)^2}}$ *and* $\mathsf{MSD}_{n,\ell,t}$ *for some* $t \in [0,1]$.

## 4 REPRESENTATIONAL STRENGTH OF TRANSFORMERS

So far we have seen multiple problems (OV, Min-IP, Max-IP and variants of MSD, LSD) that require quadratic time to solve under certain parameters assuming SETH or OVC. In this section, we show that OV and decision versions of MSD, LSD can be solved by a transformer with one attention unit in one layer.

Notice that fine-grained reduction does not trivially apply in representational strength of transformers, i.e. two problems might be subquadratic equivalent, but one problem solvable by transformers might not imply that the other one is also solvable by transformers. This is because many techniques that are simple to do on the word-RAM model (where one can do arithmetic operations, find the maximum/minimum over $n$ numbers in constant time) might not be easy to implement in parallel architectures like transformers.

We show that transformers are able to solve OV with appropriate parameters. The constructions for Min-IP, Max-IP, MSD, LSD are more complicated but follow similar ideas, so we leave them to Appendix C.

**Theorem 4.1.** *An attention unit with input and output MLPs with parameters* $d = \ell, d_{\text{in}} = \ell, d_{\text{out}} = 1, m \geq \ell + 1$ *can solve* $\mathsf{OV}_{n,\ell}$.

*Proof.* Let $v_1, \ldots, v_n \in \{0,1\}^\ell$ be an $\mathsf{OV}_{n,\ell}$ instance; define $v_{n+1} := 0^\ell$ and $X \in \mathbb{R}^{(n+1)\times\ell}$ such that $X_{i,:} = v_i$ for all $i$. Since $m \geq \ell + 1$, let $Q, K$ be arbitrary matrices such that $QK^\top = -3\log n \cdot I_\ell$ and $V \in \mathbb{R}^{\ell \times 1}$ be the all-one matrix. Let $A_{Q,K,V}$ denote this attention head, and $\varphi_1$ be the identity function.

Let $X$ be the input to the transformer with $A_{Q,K,V}$ as the only attention head. We claim that if there exists a pair $1 \leq i \neq j \leq n$ such that $\langle v_i, v_j \rangle = 0$, then $A_{Q,K,V}(X)$ has an entry which is at least $\frac{1}{n+1}$, and otherwise all entries will be at most $\frac{1}{(n+1)^{1.5}}$. As a result, we can use the second MLP $\varphi_2$ to map $A_{Q,K,V}(X)$ to 1 if any of its entry is at least $\frac{1}{n+1}$ and 0 otherwise (see Lemma D.1 for a formal proof).

We can calculate that the $i$-th entry of $A_{Q,K,V}(X)$ is

$$\sum_{j=1}^n \frac{\exp(-3\log n \cdot \langle v_i, v_j \rangle)}{\sum_{k=1}^n \exp(-3\log n \cdot \langle v_i, v_k \rangle) + 1} \cdot \|v_j\|_1 = \frac{\sum_{j=1}^n n^{-3\langle v_i, v_j \rangle}}{\sum_{k=1}^n n^{-3\langle v_i, v_k \rangle} + 1} \cdot \|v_j\|_1.$$

As a result, if there exists a pair $\langle v_{i^*}, v_{j^*} \rangle = 0$ with $1 \leq i^*, j^* \leq n$, then the $i^*$-th entry of $A_{Q,K,V}(X)$ can be lower bounded as

$$\frac{\sum_{j=1}^n n^{-3\langle v_i, v_j \rangle}}{\sum_{k=1}^n n^{-3\langle v_i, v_k \rangle} + 1} \geq \frac{n^{-3\langle v_{i^*}, v_{j^*} \rangle}}{n+1} = \frac{1}{n+1}.$$

Otherwise, $\langle v_i, v_j \rangle \geq 1$ for all $i, j$ and thus the $i$-th entry of $A_{Q,K,V}(X)$ is

$$\frac{\sum_{j=1}^n n^{-3\langle v_i, v_j \rangle}}{\sum_{k=1}^n n^{-3\langle v_i, v_k \rangle} + 1} \cdot \|v_j\|_1 \leq \sum_{j=1}^n \frac{d}{n^{3\langle v_i, v_j \rangle}} \leq n \cdot \frac{d}{n^3} < \frac{1}{n^{1.5}}.$$

$\square$

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

## A  A DETAILED INTRODUCTION TO FINE-GRAINED COMPLEXITY

### A.1  HARDNESS CONJECTURES: SETH AND OVC

We restate our central hardness conjecture here.

**Definition A.1** (Strong Exponential Time Hypothesis (SETH))**.** *For any $\varepsilon > 0$, there exists a positive integer $k$ such that $k$SAT requires $\Omega(2^{(1-\varepsilon)n})$ time.*

SETH has been one of the biggest open problem in fine-grained complexity, and one of the reasons is that SETH being true would imply $P \neq NP$. See Section 1.2 for a detailed explanation of its significance and why many people believe that it is true.

We also introduce the Orthogonal Vectors problem, which is an important problem in fine-grained complexity and will be a key intermediate problem in some of our proofs.

**Definition A.2** (Orthogonal Vectors ($\mathsf{OV}_{n,\ell}$)). *Given binary vectors $v_1, \ldots, v_n \in \{0,1\}^\ell$, $\mathsf{OV}_{n,\ell}$ asks to determine if there exists a pair $i \neq j$ such that $\langle v_i, v_j \rangle = 0$.*

Usually researchers focus on the regime when $\ell \ll n$, and thus we will assume that $\ell \leq n^{o(1)}$ throughout this paper. The straightforward algorithm for $\mathsf{OV}_{n,\ell}$ runs in time $O(n^2 \ell)$ by simply computing the inner products between each pair of vectors. In very low dimensions, one can employ a folklore recursive approach to obtain a $O(2^\ell + n)$ time algorithm (see CST (2017)). When $\ell = c \log n$ for a large constant $c$, all algorithms mentioned above require quadratic time with respect to $n$, but Abboud et al. (2015b); Chan & Williams (2021) gave a slight improvement, showing that for a fixed constant $c > 0$, $\mathsf{OV}_{n,c \log n}$ can be solved in time $n^{2-1/O(\log c)}$. This is a truly subquadratic running time for any fixed constant $c$, but becomes quadratic as $c$ grows. It is still unknown whether there exists a $O(n^{2-\varepsilon})$ time algorithm for all constant $c$, where $\varepsilon$ is an absolute constant that does not depend on $c$, and the popular OVC Conjecture states that no such algorithm exists.

**Conjecture A.3** (OVC). *For any $\varepsilon > 0$, there exists a constant $c > 0$ such that $\mathsf{OV}_{n,c \log n}$ cannot be solved in $O(n^{2-\varepsilon})$ time.*

Williams (2005) showed that assuming SETH is true, then OVC is true (the other direction is unknown). Our paper will use these two conjectures interchangeably such that our hardness results can be obtained from either conjecture.

There is also a bichromatic version of $\mathsf{OV}_{n,\ell}$ where one is given two sets of vectors $A = \{a_1, \ldots, a_n\}, B = \{b_1, \ldots, b_n\}$ such that $a_i, b_j \in \{0,1\}^\ell$ and wants to determine if there exists $i, j$ such that $\langle a_i, b_j \rangle = 0$. In fact, these two problems are subquadratic equivalent.

**Lemma A.4.** *There exists an algorithm for $\mathsf{OV}_{n,\ell}$ that runs in time $O(n^{2-\varepsilon} \cdot \mathrm{poly}(\ell))$ for some $\varepsilon > 0$ if and only if there exists an algorithm for bichromatic $\mathsf{OV}_{n,\ell}$ that runs in time $O(n^{2-\varepsilon'} \cdot \mathrm{poly}(\ell))$ for some $\varepsilon' > 0$.*

*Proof.* First we assume that there exists an algorithm $\mathcal{A}$ for bichromatic $\mathsf{OV}_{n,\ell}$ that runs in time $O(n^{2-\varepsilon} \cdot \mathrm{poly}(\ell))$ for some $\varepsilon > 0$. Given an $\mathsf{OV}_{n,\ell}$ instance $V = \{v_1, \ldots, v_n\}$ with $v_i \in \{0,1\}^\ell$ for all $i$, we first check if any $v_i$ is the all zero vector (if so then it is a yes instance). Otherwise, let $A = B = V$ and run $\mathcal{A}$ on $A, B$. Notice that since there is no all zero vector, $\langle v_i, v_i \rangle \neq 0$ for all $i$, and therefore there exists $i \neq j$ such that $\langle v_i, v_j \rangle = 0$ if and only if there exists $v_i = a \in A, v_j = b \in B$ such that $\langle a, b \rangle = 0$. The total running time is $O(n^{2-\varepsilon} \cdot \mathrm{poly}(\ell) + n\ell) = O(n^{2-\varepsilon} \cdot \mathrm{poly}(\ell))$.

Now we assume that there exists an algorithm $\mathcal{A}'$ for $\mathsf{OV}_{n,\ell}$ that runs in time $O(n^{2-\varepsilon} \cdot \mathrm{poly}(\ell))$ for some $\varepsilon > 0$. Let $A = \{a_1, \ldots, a_n\}, B = \{b_1, \ldots, b_n\}$ with vectors $a_i, b_j \in \{0,1\}^\ell$ be a bichromatic $\mathsf{OV}_{n,\ell}$ instance. For each $a_i$ we construct $a_i' = (a_i, 1, 0) \in \mathbb{R}^{\ell+2}$ and for each $b_j$ we construct $b_j' = (b_j, 0, 1) \in \mathbb{R}^{\ell+2}$ and let $A' = \{a_1' \ldots, a_n'\}, B' = \{b_1', \ldots, b_n'\}$. Notice that now $\langle a_i', a_j' \rangle, \langle b_i', b_j' \rangle \geq 1$ for all $i, j$. As a result, if there exists $\langle a_i, b_j \rangle = 0$, then $\langle a_i', b_j' \rangle = \langle a_i, b_j \rangle + 0 = 0$. Conversely, if $\langle v, w \rangle = 0$ for any $v, w \in A' \cup B'$, then it must be the case that $v = a_i'$ and $w = b_j'$ for some $i, j$ or vice versa, which implies that $\langle a_i, b_j \rangle = 0$. Therefore, running $\mathcal{A}'$ on $A' \cup B'$ will tell us whether $A, B$ is a yes instance or not. The total running time required is $O((2n)^{2-\varepsilon} \cdot \mathrm{poly}(\ell + 2)) = O(n^{2-\varepsilon} \mathrm{poly}(\ell))$. $\square$

When $\ell = c \log n$ for some constant $c > 0$, an algorithm running in $O(n^{2-\varepsilon} \cdot \mathrm{poly}(\ell))$ time for some fixed constant $\varepsilon > 0$ is a truly subquadratic algorithm. Therefore, Lemma A.4 implies that assuming OVC, there is no truly subquadratic algorithm for bichromatic $\mathsf{OV}_{n,\ell}$.

## A.2 MINIMUM INNER PRODUCT

In this section we introduce the minimum inner product problem, an important problem related to similarity search.

**Definition A.5** (Min-IP). *Given a set of binary vectors $v_1, \ldots, v_n \in \{0,1\}^\ell$, $\mathsf{Min\text{-}IP}_{n,\ell}$ asks to find one pair of $1 \leq i, j \leq n, i \neq j$ such that $\langle v_i, v_j \rangle$ is minimum.*

The trivial algorithm for $\mathsf{Min\text{-}IP}_{n,\ell}$ takes $O(n^2 \ell)$ time by enumerating all possible pairs of inner product. Sometimes we are happy with finding a pair of vectors whose inner product is close enough to optimal, so we also introduce the approximate Min-IP problem as follows.

**Definition A.6** ($\gamma$-Min-IP). *Given a set of binary vectors $v_1, \ldots, v_n \in \{0,1\}^\ell$, $\gamma$-Min-IP$_{n,\ell}$ asks to find one pair of $1 \leq i, j \leq n, i \neq j$ such that $\langle v_i, v_j \rangle$ is a $\gamma$-approximation of the minimal inner product.*

It is not hard to see that Min-IP and $\gamma$-Min-IP are both at least as hard as OV for any $\gamma \geq 1$ (just find the minimum inner product and see if it is $0$, and any multiplicative approximation of $0$ must be $0$). Therefore, assuming OVC, for any $\varepsilon > 0$ there exists $c > 0$ such that Min-IP$_{n, c \log n}$ cannot be solved in $O(n^{2-\varepsilon})$ time.

In addition, there is a decision version of Min-IP, by which we denote Min-IP$_{n,\ell,t}$, where one wants to know whether there exists a pair of vectors whose inner product is at most $t$ for some $0 \leq t \leq \ell$.

**Definition A.7** (Min-IP decision version). *Given a set of binary vectors $v_1, \ldots, v_n \in \{0,1\}^\ell$ and $0 \leq t \leq \ell$, Min-IP$_{n,\ell,t}$ asks to determine if there exists one pair of $1 \leq i, j \leq n, i \neq j$ such that $\langle v_i, v_j \rangle \leq t$.*

Min-IP$_{n,\ell}$ is trivially at least as hard as Min-IP$_{n,\ell,t}$ for any $t$ because if we can calculate the minimum, then we can decide whether it is at most $t$ or not. In addition, notice that when $t \geq \ell + 1$ or $t < 0$ the problem is trivial and requires constant time to respond.

The bichromatic versions of these problems can be defined analogously: given two sets $A, B$ with vectors in $\{0,1\}^\ell$, one needs to find $i, j$ that achieves (for bichromatic Min-IP$_{n,\ell}$) or approximates (for bichromatic $\gamma$-Min-IP$_{n,\ell}$) the minimal $\langle a_i, b_j \rangle$. One can easily obtain a truly subquadratic algorithm for all three problems above given a truly subquadratic algorithm for their bichromatic versions.

**Lemma A.8.** *Suppose there exists an algorithm $\mathcal{A}$ for bichromatic Min-IP$_{n,\ell}$ that runs in time $O(n^{2-\varepsilon} \cdot \mathrm{poly}(\ell))$ for any $\varepsilon > 0$, then there exists an algorithm for Min-IP$_{n,\ell}$ that runs in time $O(n^{2-\varepsilon'} \cdot \mathrm{poly}(\ell))$ for some $\varepsilon' > 0$. The same statement is true for $\gamma$-Min-IP$_{n,\ell}$ and Min-IP$_{n,\ell,t}$.*

*Proof.* Given a Min-IP$_{n,\ell}$ instance $V = \{v_1, \ldots, v_n\}$, we first partition $V$ into $V_1, V_2$ of equal size and run $\mathcal{A}$ on $V_1, V_2$. This will allow us to find the minimal pair in $V_1 \times V_2$. Now we further partition $V_1$ into two sets of equal size and recurse on $V_1$ to eventually find the minimal pair in $V_1 \times V_1$. Similarly we recurse on $V_2$ to find the minimal pair in $V_2 \times V_2$. The running time of our algorithm is

$$\mathrm{poly}(\ell) \cdot \sum_{i=1}^{\log n} O\left(\left(\frac{n}{2^i}\right)^{(2-\varepsilon)}\right) = O(n^{2-\varepsilon} \cdot \mathrm{poly}(\ell) \cdot \log n) \leq O(n^{2-\varepsilon'} \cdot \mathrm{poly}(\ell))$$

for some $\varepsilon' > 0$. The exact same argument holds for $\gamma$-Min-IP$_{n,\ell}$ as well because the minimal pair must appear in some recursion where we run our algorithm on. The argument also holds for Min-IP$_{n,\ell,t}$ because we have gone over all possible pairs of vectors. $\square$

## A.3  MAXIMUM INNER PRODUCT

In this section we introduce the maximum inner product problem and its variants. The problems are basically the same as problems in the previous section.

**Definition A.9** (Max-IP). *Given a set of binary vectors $v_1, \ldots, v_n \in \{0,1\}^\ell$, Max-IP$_{n,\ell}$ asks to find one pair of $1 \leq i, j \leq n, i \neq j$ such that $\langle v_i, v_j \rangle$ is maximal.*

**Definition A.10** ($\gamma$-Max-IP). *Given a set of binary vectors $v_1, \ldots, v_n \in \{0,1\}^\ell$, $\gamma$-Max-IP$_{n,\ell}$ asks to find one pair of $1 \leq i, j \leq n, i \neq j$ such that $\langle v_i, v_j \rangle$ is a $\gamma$-approximation of the maximal inner product.*

**Definition A.11** (Max-IP decision version). *Given a set of binary vectors $v_1, \ldots, v_n \in \{0,1\}^\ell$, Max-IP$_{n,\ell,t}$ asks to determine if there exists one pair of $1 \leq i, j \leq n, i \neq j$ such that $\langle v_i, v_j \rangle \geq t$.*

The bichromatic versions of these problems can be defined analogously. Similar to Min-IP, one can easily obtain a truly subquadratic algorithm for all three problems above given a truly subquadratic algorithm for their bichromatic versions again using the proof of Lemma A.8.

**Lemma A.12.** *Suppose there exists an algorithm $\mathcal{A}$ for bichromatic $(\gamma\text{-})$Max-IP$_{n,\ell}$ that runs in time $O(n^{2-\varepsilon} \cdot \mathrm{poly}(\ell))$ for any $\varepsilon > 0$, then there exists an algorithm for $(\gamma\text{-})$Max-IP$_{n,\ell}$ that runs in time $O(n^{2-\varepsilon'} \cdot \mathrm{poly}(\ell))$ for some $\varepsilon' > 0$.*

It is less obvious whether Max-IP is a harder problem than OV or not. The answer is positive, and for bichromatic Max-IP, there exists a simple proof.

**Lemma A.13.** *Suppose there exists an algorithm $\mathcal{A}$ for bichromatic Max-IP$_{n,\ell}$ that runs in time $O(n^{2-\varepsilon} \cdot \mathrm{poly}(\ell))$ for any $\varepsilon > 0$, then there exists an algorithm for OV$_{n,\ell}$ that runs in time $O(n^{2-\varepsilon} \cdot \mathrm{poly}(\ell))$.*

*Proof.* Given an OV$_{n,\ell}$ instance $v_1, \ldots, v_n \in \{0,1\}^\ell$, we partition all $v_i$ into subsets $S_1, \ldots, S_\ell$ such that each $S_j$ contains all vectors with $j$ ones. Now for each pair of $1 \leq i, j \leq \ell$, let $\bar{S}_j$ consists of vectors in $S_j$ but with all the entries flipped. As a result, for any $v \in S_i, w \in \bar{S}_j$ we have $\langle v, \bar{w} \rangle = i - \langle v, w \rangle$, which implies that $\langle v, \bar{w} \rangle = 0$ if and only if $\langle v, w \rangle = i$. Therefore, running $\mathcal{A}$ on $S_i, \bar{S}_j$ will tell us whether there is an orthogonal pair of vectors in $S_i$ and $S_j$. The total running time is $O(\ell^2 \cdot n^{2-\varepsilon} \cdot \mathrm{poly}(\ell)) = O(n^{2-\varepsilon}) \cdot \mathrm{poly}(\ell)$. $\qquad\square$

In fact, Karthik & Manurangsi (2020) proved a stronger statement which says that even approximate Max-IP$_{n,\ell}$ is stronger than OV$_{n,\ell}$ for some approximation factor.

## A.4   LEAST SIMILAR DOCUMENTS

We now formally define LSD variants, which are defined similarly to Min-IP variants. Recall that MSD and variants were defined in Section 2.3 above.

**Definition A.14** (LSD). *Given $n$ document embeddings $v_1, \ldots, v_n \in \{0,1\}^\ell$, LSD$_{n,\ell}$ asks to find $1 \leq i, j \leq n, i \neq j$ such that $\frac{\langle v_i, v_j \rangle}{\|v_i\| \cdot \|v_j\|}$ is the minimum.*

**Definition A.15** ($\gamma$-LSD). *Given $n$ document embeddings $v_1, \ldots, v_n \in \{0,1\}^\ell$, $\gamma$-MSD$_{n,\ell}$ asks to find $1 \leq i^*, j^* \leq n, i^* \neq j^*$ such that*

$$\min_{1 \leq i,j \leq n} \frac{\langle v_i, v_j \rangle}{\|v_i\| \cdot \|v_j\|} \leq \frac{\langle v_{i^*}, v_{j^*} \rangle}{\|v_{i^*}\| \cdot \|v_{j^*}\|} \leq \gamma \cdot \min_{1 \leq i,j \leq n} \frac{\langle v_i, v_j \rangle}{\|v_i\| \cdot \|v_j\|}.$$

**Definition A.16** (LSD decision version). *Given $n$ document embeddings $v_1, \ldots, v_n \in \{0,1\}^\ell$, LSD$_{n,\ell,t}$ asks to determine if there exists $1 \leq i, j \leq n, i \neq j$ such that $\frac{\langle v_i, v_j \rangle}{\|v_i\| \cdot \|v_j\|} \leq t$.*

$\gamma$-LSD$_{n,\ell}$ and LSD$_{n,\ell,t}$ are both easier than LSD. In addition, the existence of truly subquadratic time algorithm for LSD$_{n,\ell,t}$ for all $t$ again implies a truly subquadratic time algorithm for LSD$_{n,\ell}$ using binary search. Bichromatic versions of these problems can be defined analogously and the proof of Lemma A.8 again tells us that bichromatic versions are harder.

## B   PROOF OF THEOREM 3.3

In this section we prove Theorem 3.3, and our ideas are similar to the ideas in section 5 and 6 of Karthik & Manurangsi (2020). Recall the theorem as follows.

**Theorem B.1** (Theorem 3.3). *Assuming SETH or OVC and $\gamma \leq (1 + \frac{1}{\log\log n})^{\frac{\log n}{(\log\log n)^2}} = 2^{(\log n)^{1-o(1)}}$, for every $\varepsilon > 0$ there exists a constant $c > 0$ such that there is no algorithm for $\gamma$-MSD$_{n,\ell}$ that runs in time $O(n^{2-\varepsilon})$ where $\ell = (\log n)^{\frac{c \log n}{(\log\log n)^2}}$.*

We break down the proof into several lemmas below.

**Lemma B.2.** *Suppose there exists an algorithm for $\gamma$-MSD$_{n,\ell}$ where $\ell = (\log n)^{\frac{c \log n}{(\log\log n)^2}}$ for any constant $c > 0$, $\gamma \leq (1 + \frac{1}{\log\log n})^{\frac{\log n}{(\log\log n)^2}}$ that runs in $O(n^{2-\varepsilon})$ time for any $\varepsilon > 0$, then there exists an algorithm for $(1 + \frac{1}{\log\log n})$-MSD$_{n,(\log n)^k}$ with running time $O(n^{2-\varepsilon})$ for any constant $k > 0$.*

*Proof.* Let $v_1, \ldots, v_n \in \{0,1\}^{(\log n)^k}$ be an instance of $(1 + \frac{1}{\log \log n})$-MSD$_{n,(\log n)^k}$ for any constant $k > 0$. Construct $V' = \{v_1^{\otimes q}, \ldots, v_n^{\otimes q}\}$ for $q = \frac{\log n}{(\log \log n)^2}$: all the vectors in $V'$ have dimension $(\log n)^{kq} = (\log n)^{\frac{k \log n}{(\log \log n)^2}}$, so we can run the algorithm provided to find a $\gamma$-approximation of MSD on $V'$ for some $\gamma \leq (1 + \frac{1}{\log \log n})^q$. Notice that we have

$$\frac{\langle v_i^{\otimes q}, v_j^{\otimes q} \rangle}{\|v_i^{\otimes q}\| \cdot \|v_j^{\otimes q}\|} = \left( \frac{\langle v_i, v_j \rangle}{\|v_i\| \cdot \|v_j\|} \right)^q$$

for any $i, j$, so an $\gamma$-approximation of MSD on $V'$ is a $\gamma^{1/q} = (1 + \frac{1}{\log \log n})$ approximation of MSD on $v_1, \ldots, v_n$. $\qquad \square$

Now we define the bichromatic $\gamma$-Additive-MSD problem as: given a $A, B \subseteq \{0,1\}^\ell, \alpha \in [0,1]$ with $|A| = |B| = n$, we want to distinguish between the following two cases:

1. Yes instance: There exists $(a,b) \in A \times B$ such that $\frac{\langle a,b \rangle}{\|a\| \cdot \|b\|} \geq \alpha$.

2. No instance: For every $(a,b) \in A \times B$ we have $\frac{\langle a,b \rangle}{\|a\| \cdot \|b\|} < \alpha - \gamma$.

**Lemma B.3.** *Suppose there exists an algorithm $\mathcal{A}$ for $(1 + \frac{1}{\log \log n})$-MSD$_{n,(\log n)^k}$ for any constant $k > 0$ with $O(n^{2-\varepsilon})$ running time, then there exists an algorithm for bichromatic $\frac{\log n}{\ell}$-Additive-MSD$_{n,c \log n}$ for any $c > 0$ in time $O(n^{2-\varepsilon'})$ for some $\varepsilon' > 0$.*

*Proof.* The proof follows from Theorem B.4 and Theorem B.5 $\qquad \square$

**Theorem B.4** (Karthik & Manurangsi (2020) Theorem 6.2). *Suppose there exists an algorithm for $(1 + \frac{1}{\log \log n})$-MSD$_{n,(\log n)^k}$ for any constant $k > 0$ that runs in $O(n^{2-\varepsilon})$ time for any $\varepsilon > 0$, then there exists an algorithm for bichromatic $(\log n)$-Additive-Max-IP$_{n,c \log n}$ for any constant $c > 0$ that runs in $O(n^{2-\varepsilon'})$ time for some $\varepsilon' > 0$.*

**Lemma B.5.** *Bichromatic $\frac{\log n}{\ell}$-Additive-MSD$_{n,\ell}$ with $\ell = O(\log n)$ and bichromatic $(\log n)$-Additive-Max-IP$_{n,\ell'}$ with $\ell' = O(\log n)$ are subquadratic equivalent.*

*Proof.* Given a bichromatic $(\log n)$-Additive-Max-IP$_{n,\ell}$ instance with sets $A = \{a_1, \ldots, a_n\}, B = \{b_1, \ldots, b_n\}$ and integer $\alpha$, we construct $A' = \{a_1', \ldots, a_n'\}, B' = \{b_1', \ldots, b_n'\}$ as follows: for each $a_i$ we first attach $\ell - \|a_i\|_1$ many ones at the end and another $\ell + \|a_i\|_1$ zeros to obtain $a_i' \in \{0,1\}^{3\ell}$, and for each $b_j$ we first attach $\ell + \|b_j\|_1$ zeros at the end and another $\ell - \|b_j\|_1$ ones to obtain $b_j' \in \{0,1\}^{3\ell}$. Now all $a_i', b_j'$ have $\ell$ many ones, and $\langle a_i', b_j' \rangle = \langle a_i, b_j \rangle$ for all $i, j$ by our construction. Therefore, running our algorithm for $\frac{\log n}{\ell}$-Additive-MSD$_{n,\ell}$ on $A', B'$ and $\frac{\alpha}{\ell}$ will tell us whether:

1. there exists $(a', b') \in A' \times B'$ such that $\frac{\langle a', b' \rangle}{\|a'\| \cdot \|b'\|} \geq \frac{\alpha}{\ell}$, which is equivalent to $\langle a, b \rangle = \langle a', b' \rangle \geq \alpha$, or

2. for every $(a', b') \in A' \times B'$ we have $\frac{\langle a', b' \rangle}{\|a'\| \cdot \|b'\|} < \frac{\alpha}{\ell} - \frac{\log n}{\ell}$, which is equivalent to $\langle a, b \rangle = \langle a', b' \rangle < \alpha - \log n$ for all $a \in A, b \in B$.

The running time of our algorithm is $O(n^{2-\varepsilon} \cdot \text{poly}(3\ell)) = O(n^{2-\varepsilon} \cdot \text{poly}(\ell))$.

The reduction for the other direction is similar. Suppose we are given a bichromatic $\frac{\log n}{\ell}$-Additive-MSD$_{n,\ell}$ instance with sets $A = \{a_1, \ldots, a_n\}, B = \{b_1, \ldots, b_n\}$ and $\alpha \in [0,1]$, we construct the exact same $A', B'$ as before such that $\langle a_i', b_j' \rangle = \langle a_i, b_j \rangle$ for all $i, j$. Since $\|a_i'\| = \|b_j'\| = \sqrt{\ell}$ for all $i, j$ now, the same argument implies that running the algorithm on $A', B'$ and $\alpha \cdot \ell$ will solve the problem. $\qquad \square$

**Lemma B.6** (Chen (2020)). *Suppose there exists an algorithm for* $(\log n)$*-Additive-BMax-IP*$_{n,c\log n}$ *for any constant* $c > 0$ *running in time* $O(n^{2-\varepsilon'})$ *for some* $\varepsilon' > 0$*, then there exists an algorithm for* $\mathsf{OV}_{n,c'\log n}$ *for any constant* $c' > 0$ *with running time* $O(n^{2-\frac{\varepsilon'}{2}})$*, thus refuting* SETH *and* OVC.

*Proof of Theorem 3.3.* This follows from a combination of Lemma B.2, Lemma B.3, Lemma B.5 and Lemma B.6. □

## C  TRANSFORMERS CAN SOLVE Max-IP, Min-IP, MSD$_{n,\ell,t}$ AND LSD$_{n,\ell,t}$

**Theorem C.1.** *A attention unit with input and output MLPs with parameters* $d = \ell, d_{\text{in}} = \ell + 1, d_{\text{out}} = 1, m \geq \ell + 1$ *can solve* Max-IP$_{n,\ell,t}$ *and* Min-IP$_{n,\ell,t}$ *for* $1 \leq t \leq \ell$.

*Proof.* Given a Max-IP$_{n,\ell}$ instance $v_1, \ldots, v_n \in \{0,1\}^\ell$ and $V \in \mathbb{R}^{n \times \ell}$ such that $V_{i,:} = v_i$ for all $i$, let $x_i = (v_i, 1) \in \{0,1\}^{\ell+1}$ for all $i$, $x_{n+1} := (0, \ldots, 0, t+1) \in \mathbb{R}^{\ell+1}$ and $X \in \mathbb{R}^{(n+1) \times (\ell+1)}$ be such that $X_{i,:} = x_i$ for all $i$. Let $Q, K \in \mathbb{R}^{d_{\text{in}} \times m}$ be such that $QK^\top = 3\log n \cdot I_{d_{\text{in}}}$, $V = (1, 1, \ldots, 1, 0) \in \mathbb{R}^{\ell+1}$ and $A_{Q,K,V}$ denote this attention head.

We want to send $X$ to the attention head, which could be done by many ways given the input $V$. For example, we can add a "end token" [5] to the $n$ documents that is always embedded into the vector $(0, \ldots, 0, t+1) \in \mathbb{R}^\ell$. Then we can use a MLP to send $V$ to $X$ (see Lemma D.2 for a formal proof). Now we can check that the $i$-th entry of $A_{Q,K,V}(X)$ is

$$\sum_{j=1}^n \frac{\exp(3\log n \cdot \langle x_i, x_j \rangle)}{\sum_{k=1}^n \exp(3\log n \cdot \langle x_i, x_k \rangle) + \exp(3(t+1)\log n)} \cdot \|v_j\|_1 = \frac{\sum_{j=1}^n n^{3\langle x_i, x_j \rangle} \cdot \|v_j\|_1}{\sum_{k=1}^n n^{3\langle x_i, x_k \rangle} + n^{3(t+1)}}.$$

Let $1 \leq i^* \neq j^* \leq n$ be such that $\langle v_{i^*}, v_{j^*} \rangle$ is the largest. Therefore, if $\langle v_{i^*}, v_{j^*} \rangle \geq t$, then $\langle x_{i^*}, x_{j^*} \rangle \geq t+1$, which means that

$$\frac{\sum_{j=1}^n n^{3\langle x_{i^*}, x_j \rangle} \cdot \|x_j\|_1}{\sum_{k=1}^n n^{3\langle x_{i^*}, x_k \rangle} + n^{3(t+1)}} \geq \frac{n^{3\langle x_{i^*}, x_{j^*} \rangle} \cdot \|v_{j^*}\|_1}{\sum_{k=1}^n n^{3\langle x_{i^*}, x_k \rangle} + n^{3(t+1)}} \geq \frac{\|v_{j^*}\|_1}{n+1} \geq \frac{1}{n+1}.$$

On the other hand, if $\langle v_{i^*}, v_{j^*} \rangle \leq t-1$, then $\langle x_{i^*}, x_{j^*} \rangle \leq t$, which means

$$\frac{\sum_{j=1}^n n^{3\langle x_{i^*}, x_j \rangle} \cdot \|v_j\|_1}{\sum_{k=1}^n n^{3\langle x_{i^*}, x_k \rangle} + n^{3(t+1)}} \leq \ell \cdot \frac{n \cdot n^{3t}}{n^{3(t+1)}} = \frac{\ell}{n^2} < \frac{1}{(n+1)^{1.5}}.$$

Therefore, we can use the second MLP $\varphi_2$ to map $A_{Q,K,V}(X)$ to 1 if any of its entry is at least $\frac{1}{n+1}$ and 0 if all its entries are at most $\frac{1}{(n+1)^{1.5}}$ (see Lemma D.1 for a formal proof of existence).

The proof for Min-IP is almost the same, except that we new let $QK^\top = -3\log n \cdot I_{d_{\text{in}}}$ and $x_{n+1} = [0, \ldots, 0, -(t+1)] \in \mathbb{R}^{\ell+1}$ instead. Now the $i$-th entry of $A_{Q,K,V}(X)$ is

$$\frac{\sum_{j=1}^n n^{-3\langle x_i, x_j \rangle} \cdot \|v_j\|_1}{\sum_{k=1}^n n^{-3\langle x_i, x_k \rangle} + n^{-3(t+1)}}.$$

Let $i^*, j^*$ be such that $\langle x_{i^*}, x_{j^*} \rangle$ is the smallest. Therefore, if $\langle v_{i^*}, v_{j^*} \rangle \leq t$, then $\langle x_{i^*}, x_{j^*} \rangle \leq t+1$, which means that

$$\frac{\sum_{j=1}^n n^{-3\langle x_{i^*}, x_j \rangle} \cdot \|v_j\|_1}{\sum_{k=1}^n n^{-3\langle x_{i^*}, x_k \rangle} + n^{-3(t+1)}} \geq \frac{n^{-3\langle x_{i^*}, x_{j^*} \rangle}}{\sum_{k=1}^n n^{-3\langle x_{i^*}, x_k \rangle} + n^{-3(t+1)}} \geq \frac{1}{n+1}.$$

On the other hand, if $\langle v_{i^*}, v_{j^*} \rangle \geq t+1$, then $\langle x_{i^*}, x_{j^*} \rangle \geq t+2$, which means

$$\frac{\sum_{j=1}^n n^{-3\langle x_{i^*}, x_j \rangle} \cdot \|v_j\|_1}{\sum_{k=1}^n n^{-3\langle x_{i^*}, x_k \rangle} + n^{-3(t+1)}} \leq \ell \cdot \frac{n \cdot n^{-3(t+2)}}{n^{-3(t+1)}} = \frac{\ell}{n^2} < \frac{1}{(n+1)^{1.5}}.$$

□

---

[5]Sanford et al. (2024b) has a similar assumption.

**Theorem C.2.** *A attention unit with input and output MLPs with parameters $d = \ell, d_{\text{in}} = \ell + 1, d_{\text{out}} = 1, m \geq \ell + 1$ can solve $\mathsf{MSD}_{n,\ell,t}$ and $\mathsf{LSD}_{n,\ell,t}$ for any $t \in [0,1]$.*

*Proof.* When $t = 0$ the problem is trivial so we simply need to output $1$ using MLPs, so without losing of generality we assume $t \neq 0$. Given a $\mathsf{MSD}_{n,\ell}$ instance $v_1, \ldots, v_n \in \{0,1\}^\ell$ and $V \in \mathbb{R}^{n \times \ell}$ such that $V_{i,:} = v_i$ for all $i$, let $x_i = (\frac{v_i}{\|v_i\|_1}, 1) \in \mathbb{R}^{\ell+1}$ for all $i$, $x_{n+1} := (0, \ldots, 0, t+1) \in \mathbb{R}^{\ell+1}$ and $X \in \mathbb{R}^{(n+1) \times (\ell+1)}$ be such that $X_{i,:} = x_i$ for all $i$. Let $Q, K \in \mathbb{R}^{d_{\text{in}} \times m}$ be such that $QK^\top = 3 \log n \cdot I_{d_{\text{in}}}, V = (1, 1, \ldots, 1, 0) \in \mathbb{R}^{\ell+1}$ and $A_{Q,K,V}$ denote this attention head.

Since MSD is exactly Max-IP after we normalize the document embeddings, the proof of Theorem C.1 implies that it suffices to send $X$ to the attention head. See Lemma D.3 for a construction.

The proof for LSD is exactly the same as the proof for Min-IP after applying the $\varphi_1$ construced in Lemma D.3. $\qquad\square$

## D  MLP CONSTRUCTIONS

**Lemma D.1.** *For any $a, b \in \mathbb{R}$ such that $b > a$, there exists a continuous function $f : \mathbb{R}^\ell \to \mathbb{R}$ such that*

$$f(x) = \begin{cases} 1 & \text{if } x[i] \geq b \text{ for any } 1 \leq i \leq \ell \\ 0 & \text{if } x[i] < a \; \forall 1 \leq i \leq \ell. \end{cases}$$

*Proof.* Firstly we define $g : \mathbb{R}$ such that

$$g(x) = \begin{cases} 1 & \text{if } x \geq b \\ \frac{1}{b-a}(x-a) & \text{if } a \leq x < b \\ 0 & \text{if } x < a. \end{cases}$$

Now we let

$$f(x) = 1 - \prod_{i=1}^{\ell} (1 - g(x[i])).$$

It is not hard to see that $g$ is a continuous function, and therefore $f$ is a continuous function. When $x[i] \geq b$ for any $1 \leq i \leq \ell$, $g(x[i]) = 1$ and therefore $f(x) = 1 - 0 = 1$. On the other hand, if $x[i] < a$ for all $i$, then $f(x) = 1 - 1 = 0$. $\qquad\square$

**Lemma D.2.** *There exists a continuous function $f : \mathbb{R}^\ell \to \mathbb{R}^{\ell+1}$ such that*

$$f(x) = \begin{cases} (x, 1) & \text{if } x[\ell] \leq 1 \\ (0, x) & \text{otherwise.} \end{cases}$$

*Proof.* First we define a function $g : \mathbb{R} \to \mathbb{R}$ such that

$$g(x) = \begin{cases} 1 & \text{if } x \leq 1 \\ 2 - x & \text{if } 1 < x \leq 2 \\ 0 & \text{if } x > 2, \end{cases}$$

and we also define $f_1, f_2 : \mathbb{R}^\ell \to \mathbb{R}^{\ell+1}$ where

$$f_1(x) = (x, 1), f_2(x) = (0, x).$$

It is not hard to see that $g, f_1, f_2$ are all continuous functions, so we let

$$f(x) = g(x[\ell]) \cdot f_1(x) + (1 - g(x[\ell])) \cdot f_2(x)$$

such that $f$ is also continuous. We can check that $f$ satisfies the requirement in the lemma statement. $\qquad\square$

**Lemma D.3.** *There exists a continuous function $f : \mathbb{R}^\ell \to \mathbb{R}^{\ell+1}$ such that*

$$f(x) = \begin{cases} (\frac{x}{\|x\|_1}, 1) & \text{if } x[d] \leq 1 \\ (0, \frac{x}{\|x\|_1}) & \text{otherwise.} \end{cases}$$

*Proof.* Let $g$ be the same functions as in Lemma D.2 and $f_1, f_2 : \mathbb{R}^\ell \to \mathbb{R}^{\ell+1}$ where

$$f_1(x) = \left( \frac{x}{\|x\|_1}, 1 \right), f_2(x) = \left( 0, \frac{x}{\|x\|_1} \right).$$

$f_1, f_2$ are continuous function over $\mathbb{R}^\ell$, and therefore we let

$$f(x) = g(x[\ell]) \cdot f_1(x) + (1 - g(x[\ell])) \cdot f_2(x)$$

such that $f$ is also continuous. We can check that $f$ satisfies the requirement in the lemma statement.

$\square$

