# OpenReview forum: "Fundamental Limitations on Subquadratic Alternatives to Transformers"
_ICLR.cc/2025/Conference — ICLR 2025 Poster_

### Official Review · Reviewer_32nU · 2024-11-03

**Soundness:** 3
**Presentation:** 4
**Contribution:** 2
**Rating:** 6
**Confidence:** 3

**Summary:**

Paper is well written and brings interesting results from complexity theory to the deep learning community. It describes theoretical  limitations of sub-quadratic attention architectures for similarity and related problems. Paper shows that MSD, LSD and their variants require quadratic time assuming SETH and therefore any subquadratic alternatives to transformers are not able to solve them due to computational constraints. Also the paper explicitly constructs one head / one layer transformer which is able to solve such a problem, therefore making clear an important difference between standard attention and subquadratic alternatives.

**Strengths:**

1. General results on limitations on sub-quadratic  architecture or heuristic for most similar, least similar and related problems. Proved that accuracy loss for any task relating to document similarity is unavoidable for any sub-quadratic approach.
2. Shown that one head / one layer transformer can solve those tasks.  This includes explicit transformer construction.
3. Bringing complexity results (Strong Exponential Time Hypothesis) to transformer architectures.

**Weaknesses:**

1. It is not exactly clear when the paper talks about approximate or exact results.
2. It relies on the BOW (bag of words) model, which already introduces approximation in practical settings. This makes results (which are about exact solution) weaker.
3. Considering that finding closest pairs in Euclidean or Manhattan distance both require quadratic time (assuming SETH) is well know, results for MSD, LSD are fairly incremental.

**Questions:**

> We focus on document similarity tasks, where one is given
as input many documents and would like to find a pair which is (approximately)
the most similar.

Could you please elaborate, why 'approximately' mentioned here? Is it not SETH and derived conjectures about exact results only?

> In other words, in order to perform tasks that (implicitly or explicitly) involve document similarity, one may as well use Transformer and cannot avoid its quadratic running time.

Approximate similarity search involves tradeoff precision and run time. If we talk about the exact solution of the similarity search, it would be nice to make it explicit.

> bag-of-words embeddings
How important is usage of bag of words embedding?  Would it be possible to use embedding / technique which takes into account word ?

---

> ### Author Response · Authors · 2024-11-14
> **Rebuttal by Authors**
>
> We thank the reviewer for the thoughtful comments and questions.
>
> **Approximation results:** In addition to proving the hardness of exact document similarity, we also proved in this paper the hardness of approximate document similarity. We proved that approximation versions of MSD, LSD (which we denoted by $\gamma$-MSD, $\gamma$-LSD) also require quadratic time in our main theorems (Theorem 1.1 and 1.2). Here, approximation means we only need to find one pair of documents whose distance is at most $\gamma$ times the distance of the optimal pair for LSD and at least $1/\gamma$ times the distance of the optimal pair for MSD. As a result, they are also hard for subquadratic models, and this is why we refer to approximation versions throughout the abstract and introduction.
>
> **SETH and approximation:** Indeed, SETH conjectures the hardness of an exact problem, but it has since been used many times in the literature to prove hardness of approximation problems as well. For example, we cite the papers [1,2,3] which all proved hardness of approximation problems, including closest pair, Max-IP etc, from SETH. We used these results to prove our own hardness of approximate document similarity.
>
> **Bag-of-Words embedding:** As mentioned above, we also show hardness of approximation, which could mitigate the impact of an embedding like bag-of-words not completely capturing the document. That said, our results hold for many other embeddings. For example, they apply almost directly to TF-IDF without needing to modify the proofs much.
>
> **Simple reductions:** You’re right that many of our reductions are fairly simple, but we view this as a merit: it means that the practical hardness of problems like SAT, OV, etc translate directly to giving practical hardness here as well. Complicated reductions could otherwise make instances become impractical, but we avoid that here.
>
> [1] C. S. Karthik and Pasin Manurangsi. On closest pair in euclidean metric: Monochromatic is as
> hard as bichromatic. Combinatorica, 40(4):539–573, aug 2020. ISSN 0209-9683. doi: 10.1007/
> s00493-019-4113-1.
>
> [2] Lijie Chen. On the hardness of approximate and exact (bichromatic) maximum inner product. Theory Of Computing, 16(4):1–50, 2020.
>
> [3] Aviad Rubinstein. Hardness of approximate nearest neighbor search. In Proceedings of the 50th
> Annual ACM SIGACT Symposium on Theory of Computing, STOC 2018, pp. 1260–1268, New
> York, NY, USA, 2018. Association for Computing Machinery. ISBN 9781450355599. doi: 10.
> 1145/3188745.3188916.

---

> ### Author Response · Authors · 2024-11-24
> **Reminder**
>
> A gentle reminder that the discussion period is going to close soon. Please let us know if there are any questions, thanks!

---

> > ### Comment · Reviewer_32nU · 2024-11-25
> >
> > Thank you for clarification and references

---

### Official Review · Reviewer_8QVS · 2024-11-03

**Soundness:** 3
**Presentation:** 3
**Contribution:** 4
**Rating:** 8
**Confidence:** 5

**Summary:**

Attention is a computationally bottleneck of the prominent transformer models used for language modelling. The classic variant is in quadratic (on the input-dimension) complexity class, which is why sub-quadratic variants such as state-space models emerged, to enable e.g. longer document processing.
The authors claim that subquadratic alternatives to transformers face inherent limitations in performing specific NLP tasks, namely document similarity. Its contributions are theoretically based on complexity theory; namely, 1) the prominent SETH problam cannot be solved sub-quadratic, and 2) a constructed transformer can solve SETH.

**Strengths:**

Finally some more mathematically theoretically founded analysis of the prominent architecture and its limitations.
It analysis seems rigorous and its offering insights that can influence future research on alternative architectures in NLP.

**Weaknesses:**

minors/ missing discussions:

1.1) The paper would benefit from discussing the practical impacts of its findings. I.p. the bounds of its proof seem to be quite practically relevant and not 'entirely asymptotic' - in praxis it could be more relevant to have 'bad asymptotic with good bounds'.

1.2) It would be beneficial to explore the performance of alternative architectures i.p. w.r.t. practicability - i am not sure if a state space machine can't handle OVC in a reasonable depth like 10 layers.

1.3) it would be beneficial to discuss a broader range of tasks

mediocores:

2.1) i find the main paper pretty hard to read and would advice a bit of restructering i.p.
- abstract line 22ff are quite redundant
- intro line 132ff seem pretty random/ not needed/ more distractive
- repetitively bichromatic versions are mentioned but not required in the core eg 2.2.1, 2.2.2, 2.3 -> i would love to see that in a final/following discussion sections once. Similarly Min/Max-IP are pretty confusing as not required for the main result -> discussion afterwards
- you define in 2.3 MSD, but actually require LSD for the core proof. that is one part that actually should be written explicitly twice (or the other version) :)

2.2) the core result is 4. it, and its impact should be discussed further, in particular recall what sub-quadratic methods suffer from.
the proof could use a rewrite. i.p.: 4.1. is a transformer from your definition (not just attention~...). i suffer a bit to understand the step 529: A_{Q,K,V}(X) IS NOT line 528, but it is upperbounded by it which is all you need (?) you miss the factors XV = in this construction #ones in v_i.. or did i miss something?

**Questions:**

- 259: finds
- 310: multi-layer
- 434 write the numbers explicitly, 'very large' is random here
- 534 left side v_i^*  (the star)
- line 529 as mentioned above

---

> ### Author Response · Authors · 2024-11-14
> **Rebuttal by Authors**
>
> We thank the reviewer for the insightful comments.
>
> **Practical impacts:** Indeed, most of our hardness results come from simple reductions. For example, if we have an algorithm for LSD that runs in time $T$, then that gives us an algorithm for OV that runs in time at most $2T$. Thus, the practical hardness of problems like SAT and OV carries over to yield practical hardness here as well. We will expand on this in the final version.
>
> **Performance of other models solving OV:** Indeed, it could be interesting to see how to pick the depth for certain subquadratic models like SSMs to solve document similarity tasks. Since our results show that these tasks need quadratic time, one would need to pick a prohibitively large depth for larger $n$ which scales linearly with $n$ (to get a final quadratic running time).
>
> **Broader range of tasks:** We aimed to study a wide variety of similarity search and document similarity problems in this paper (exact vs approximate, monochromatic vs bichromatic, closest vs farthest, optimization vs decision, etc), which appear (either explicitly or implicitly) in many different tasks that language models are intended to solve. That said, we fully agree that finding more examples and other types of tasks would be exciting.
>
> **Other writing suggestions:** Thank you, we agree with all the suggestions and corrections you made and will update them in the final version. In particular, for line 528 in the proof of Theorem 4.1, you’re right that we are missing the $\ell_1$ norm of $v_j$ in the summation, although it does not impact the correctness of the proof since it is upper bounded by $d$, which is much smaller than the multiplicative gap of $\sqrt{n}$.

---

> ### Author Response · Authors · 2024-11-24
> **Reminder**
>
> A gentle reminder that the discussion period is going to close soon. Please let us know if there are any questions, thanks!

---

> ### Comment · Reviewer_8QVS · 2024-11-25
>
> Thank you for replying, and addressing my concerns.
>
> I still find my rating solid.
> For increasing it further I would require a thorough discussion on the practicality of your bounds found in the proof (point 1.1) and what i flagged in minors, w.r.t. standard model hyperparameters (sequence lengths). I found your one-liner not sufficient here J (and also not the other review comments).
> I actually wonder why you do not interpret the bounds further, from first reading the paper, I thought they were quite close to being practically relevant as well.
>
> greets

---

> > ### Author Response · Authors · 2024-11-26
> > **Response from Authors**
> >
> > Thank you for your comments and engagement. Currently all the lower bounds in our paper are based on SETH and the OV Conjecture, which are two of the “gold standards” in fine-grained complexity theory. Since most of our reductions themselves are practical, any practical lower bounds for those problems translate to practical lower bounds here as well. In other words, if one breaks our lower bounds in a “practical way” (for example, by showing that there exists an algorithm for MSD with $n$ documents and embedding size $d$ that runs in time $d^{10}\cdot n^{1.1}$), then this would be a **huge** breakthrough in fine-grained complexity theory, and to name a few:
> > 1. This will **drastically** improve the current state-of-art algorithm for OV with $n$ vectors of length $c\log n$ from $n^{2-1/O(\log c)}$ (just barely better than $n^2$) to $O(n^{1.11})$ (way better than $n^2$);
> > 2. Similarly, this will **drastically** improve the current state-of-art algorithm for Max-IP with $n$ vectors of length $c\log n$ from $n^{2-O(1/\sqrt{c})}$ to $O(n^{1.11})$.
> >
> > We will add a thorough discussion on this in the final version of this paper.

---

### Official Review · Reviewer_TK7Z · 2024-11-03

**Soundness:** 2
**Presentation:** 2
**Contribution:** 2
**Rating:** 3
**Confidence:** 4

**Summary:**

The paper studies a fundamental issue whether any subquadratic approximations to the transformers' quadratic attention mechanisms can solve perfectly the minimum inner product problem (or its variants (such as the maximum inner product one)) among a set of binary vectors. Built on results about the inherent complexity of such problems, the authors show that such problems can not be solved by any subquadratic approximations due to the complexity mismatch. They also show the problems can be solved by a transformer using the quadratic attention mechanisms by constructing such a network.

**Strengths:**

The results are theoretical, demonstrating a fundamental limitation of any subquadratic approximations of the transformers' quadratic attention mechanisms. The involved steps seem sound.

**Weaknesses:**

I believe the results, while rigorous and sound, have almost no connection with the transformers being used in practice. Generally speaking, the transformers are shown to be very capable of solving different kinds of problems empirically. For example, subquadratic approximations try to show that they can perform similarly to the original transformers but more efficiently, which is orthogonal to the results in the paper. Due to the nature of the results, there are no experimental results. But the variants of the transformers being used are mainly justified via results. Furthermore, it is known that at least some of the transformers (such as the decoder-only ones) can not solve counting and copying problems [1] perfectly and the additional impact of the results in the paper to the research on transformers may be very limited.

[1] Federico Barbero, Andrea Banino, Steven Kapturowski, Dharshan Kumaran, Jo˜ao GM Ara´ujo, Alex Vitvitskyi, Razvan Pascanu, and Petar Veliˇckovi´c. Transformers need glasses! information oversquashing in language tasks. arXiv preprint arXiv:2406.04267, 2024 (also in NeurIPS 2024).

**Questions:**

1. Can the results be enhanced to rank different subquadratic approximations?
2. Similarly, can the results be enhanced to quantify the gap between the quadratic transformers and the subquadratic approximations?
3. Are there approximate algorithms for solving the minimum inner product problem or its variants and how would these relate to the subquadratic approximations?

---

> ### Author Response · Authors · 2024-11-14
> **Rebuttal by Authors**
>
> We thank the reviewer for the thoughtful review.
>
> **Empirical interpretation:** Indeed, as the reviewer explains, the papers introducing subquadratic alternatives typically justify them with experiments showing that they are empirically fast and accurate. The experiments differ in details in different papers, but roughly, they show that their new model runs faster than transformer and achieves nearly the same accuracy, typically slightly worse. (See, for instance, figure 3 in [2], figure 2 of [3]) One may wonder whether it is possible to design a subquadratic model that actually achieves the same accuracy, or whether an accuracy loss is necessary. Our results show that, for document similarity tasks, it is not possible for any subquadratic model to achieve the same accuracy as transformer, and so these accuracy losses observed in all these experimental papers must necessarily appear for document similarity tasks. In other words, our paper is giving a theoretical explanation for exactly the empirical phenomenon that the reviewer is mentioning.
>
> **Other limitations on transformers:** Indeed, in addition to [1], there has been much work studying problems that transformer is unable to solve. For instance, the papers [4] and [5] which we reference in our paper prove that transformer cannot solve problems based on triple-wise correlations or on tasks that are hard to parallelize. More in the spirit of our paper, one can also take problems which require more than quadratic time (such as problems about finding correlations in tensors that require cubic time), and these also cannot be solved by transformer.
>
> This is why it is important that we also prove transformer can solve the document similarity tasks we discuss in this paper. We are highlighting a class of problems that transformer can solve, but subquadratic models cannot, thus showing subquadratic models must be worse than transformer at these tasks.
>
> **Ranking or gaps with subquadratic alternatives:** Our results don’t show how to rank subquadratic alternatives, they only show that these alternatives cannot solve document similarity problems. Finding problems that some variants can solve and others cannot would give an interesting way to rank these alternatives, although we note that one would need to focus on problems that can be solved in linear time in order to rank alternatives which run in linear time. Using such problems to quantify the gaps between different models could also be interesting future work.
>
> **Approximation algorithms:** Unfortunately, approximating the minimum inner product is just as hard as computing it exactly. See Theorem 3.1 in our paper where we prove this. (We prove it for approximate LSD instead of approximate Min-IP, but the essentially identical proof works for approximate Min-IP as well.)
>
>
> [2] Sinong Wang, Belinda Z. Li, Madian Khabsa, Han Fang, and Hao Ma. Linformer: Self-attention
> with linear complexity. CoRR, abs/2006.04768, 2020.
>
> [3] Praneeth Kacham, Vahab Mirrokni, and Peilin Zhong. Polysketchformer: Fast transformers
> via sketching polynomial kernels. In Forty-first International Conference on Machine Learn-
> ing, ICML 2024, Vienna, Austria, July 21-27, 2024.
>
> [4] Clayton Sanford, Daniel Hsu, and Matus Telgarsky. Transformers, parallel computation, and logarithmic depth. In Ruslan Salakhutdinov, Zico Kolter, Katherine Heller, Adrian Weller, Nuria Oliver, Jonathan Scarlett, and Felix Berkenkamp (eds.), Proceedings of the 41st International Conference on Machine Learning, volume 235 of Proceedings of Machine Learning Research, pp. 43276–43327. PMLR, 21–27 Jul 2024a.
>
> [5] Clayton Sanford, Daniel Hsu, and Matus Telgarsky. Representational strengths and limitations of transformers. In Proceedings of the 37th International Conference on Neural Information Processing Systems, NIPS ’23, Red Hook, NY, USA, 2024b. Curran Associates Inc.

---

> > ### Comment · Reviewer_TK7Z · 2024-11-25
> > **Inaccurate Statement: "it is not possible for any subquadratic model to achieve the same accuracy as transformer"**
> >
> > I think the following statements "Our results show that, for document similarity tasks, it is not possible for any subquadratic model to achieve the same accuracy as transformer, and so these accuracy losses observed in all these experimental papers must necessarily appear for document similarity tasks. In other words, our paper is giving a theoretical explanation for exactly the empirical phenomenon that the reviewer is mentioning." are not supported by the results in the paper. The results in the paper show that it is not possible for subquadratic model to solve the document similarity task perfectly. When both quadratic and subquadratic models cannot solve a problem perfectly, the results in the paper do not provide a way to rank them quantitatively.

---

> > > ### Author Response · Authors · 2024-11-26
> > > **Response from Authors**
> > >
> > > Thank you for your reply. In our paper we showed that the standard (quadratic) transformers **can** solve the problem (OV, threshold versions of MSD, LSD) perfectly theoretically in section 4 while the subquadratic models **cannot** in section 3. This creates a separation between standard transformers and subquadratic models.
> > >
> > > Regarding our statement that you pointed out, these two results that we prove show that subquadratic models cannot solve these document similarity tasks to the same accuracy as standard transformers, which is supported by the papers we cited in our previous response.

---

> > ### Comment · Reviewer_TK7Z · 2024-11-25
> >
> > I have read all the reviews and the authors' replies so far. I think my original assessment is still accurate.

---

> ### Author Response · Authors · 2024-11-24
> **Reminder**
>
> A gentle reminder that the discussion period is going to close soon. Please let us know if there are any questions, thanks!

---

> ### Comment · Reviewer_8QVS · 2024-11-25
>
> Dear Reviewer TK7Z,
>
> to the best of my knowledge the ICLR mission statement is not entirely/ necessarily focused on empirical studies.
>
> Albeit I am pretty interested in the practical implications as well, I also see the merit of theoretical bounds. They definitely should be discussed as good as possible, but they may also be sharpened (or becoming more relevant) in follow up work.
>
> On the other hand you give [1] as a reference, which (to my eyes) is a pretty obvious fact, evaluated on their inhouse models only (that are not primarily trained on the evaluated tasks --- or maybe they are - who knows?), and pretty strange incompletely described toy models (C.4/6) which leaves the reader with just as many questions regarding "what is possible, still"...
>
> Thank you for reflecting on my concern.

---

### Official Review · Reviewer_97XG · 2024-11-04

**Soundness:** 2
**Presentation:** 2
**Contribution:** 3
**Rating:** 6
**Confidence:** 1

**Summary:**

This paper introduces a theoretical perspective on the ability of transformers to solve document similarity tasks. Specifically, by relying on the SETH conjecture, the paper connects complexity theory with transformers, and shows that subquadractic models cannot solve a class of problems that are solvable by transformers, which are quadractic.

**Strengths:**

Although this is not my area of expertise, I believe the paper's main findings, particularly regarding document similarity tasks, have the potential to impact the literature on architectural development. Additionally, to the best of my knowledge, the paper is well-written, the math is accurate and concise.

**Weaknesses:**

While the theoretical coverage seems accurate to me, I was disappointed by the lack of empirical evidence supporting the paper's main findings. For instance, experiments demonstrating that subquadratic models cannot solve Max-IP, Min-IP, MSD, and LSD, whereas transformers can, would have been valuable.

**Questions:**

For MSD and LSD, how relevant are your findings in practical applications, such as recommendation systems? Could varying values of sequence length $n$ and model parameters lead to different practical conclusions?

---

> ### Author Response · Authors · 2024-11-14
> **Rebuttal by Authors**
>
> We thank the reviewer for the thoughtful review.
>
> **Empirical evidence:** We agree that experiments showing particular models failing to solve document similarity tasks could be interesting in future work. However, we want to emphasize that our results, which mathematically prove that subquadratic models cannot solve document similarity, are more general than any such experiment. For instance, an experiment showing that one particular model cannot solve document similarity may not be very convincing, since one could imagine slightly changing the model to get around the issue. Our results show that it is mathematically impossible to do any such modification without using quadratic time.
>
> **Practical applications on recommendation systems:** Indeed, our work is directed related to the maximum inner product search (MIPS) problem in recommendation systems. In the MIPS problem we are given a dataset $P$ of $n$ points and a query $q$ such that the goal is to return a data point $p \in P$ such that $\langle p,q\rangle$ is the largest. In recommendation systems, $P$ could be a set of items and $q$ could be a customer, such that each time a customer comes, we want to know which item the customer likes the most. This problem is essentially equivalent to the Max-IP problem we study and prove lower bounds with in this paper. We will add another paragraph on this application in later versions.
>
> **Varying sequence length and model parameters:** In general, increasing the model parameters in terms of $n$ would make the model more powerful. However, our results show the hardness of these document similarity problems, which are defined independently of what the model parameters are (see our footnote 2 on page 4). So, one would at least need to very substantially increase the parameters, to a regime where the running time becomes quadratic, in order to overcome our lower bound.

---

> ### Author Response · Authors · 2024-11-24
> **Reminder**
>
> A gentle reminder that the discussion period is going to close soon. Please let us know if there are any questions, thanks!

---

### Meta-Review · Area_Chair_WX1c · 2024-12-26

**Metareview:**

Reviewers agree that the paper considers an important problem of understanding the limitations of subquadratic alternatives to attention. However they disagree on the utility of the complexity theory based separation results shown in this paper. In particular the relation between hardness of solving MSD in subquadratic time and any practical tasks is missing. The recommendation system example given by authors is not convincing, as large scale practical systems do not uses quadratic computation for recommendations. Overall I think the paper is on borderline with interesting research direction, but missing connections to practice. I suggest acceptance in the hope that this encourages future works that builds these connections.

**Additional Comments On Reviewer Discussion:**

Questions have been mainly around practical impact of the results, clarification on the proofs. Authors response alleviates these concerns.

---

### Decision · Program_Chairs · 2025-01-22

Accept (Poster)